# Scalable and Stable Surrogates for Flexible Classifiers with Fairness Constraints

**Harry Bendekgey**[*]
hbendekg@uci.edu

**Erik B. Sudderth**[*]
sudderth@uci.edu

## Abstract

We investigate how fairness relaxations scale to flexible classifiers like deep neural networks for images and text. We analyze an easy-to-use and robust way of imposing fairness constraints when training, and through this framework prove that some prior fairness surrogates exhibit degeneracies for non-convex models. We resolve these problems via three new surrogates: an adaptive data re-weighting, and two smooth upper-bounds that are provably more robust than some previous methods. Our surrogates perform comparably to the state-of-the-art on low-dimensional fairness benchmarks, while achieving superior accuracy and stability for more complex computer vision and natural language processing tasks.

## 1 Introduction

A growing body of research has explored the ethical and societal implications of increasingly automated decision-making. One critical concern is whether particular machine learning algorithms perform differently on average for members of marginalized groups. In addition to being an ethical question, this is a legal one: Attributes such as race and gender are legally-protected categories, and an employer whose decision-making process disproportionately hurts one group can be litigated for committing adverse or disparate impact. One quantitative notion of fairness is called *demographic parity*, which states that a decision maker should accept (or classify as positive) a roughly equal proportion of each group [8]. An employer who hires many male applicants, but few female applicants, violates demographic parity by disproportionately allocating resources to one sensitive group.

Machine learning is also increasingly used by law enforcement, where ethical questions become civil rights questions. Facial recognition and risk assessment algorithms have been shown to have disparities in accuracy between black and white defendants [2, 7], which may lead to the arrest of an innocent person [29]. The *equalized odds* [27] fairness criterion requires a decision maker to have the same false negative rates, as well as the same false positive rates, for each sensitive group. *Equality of opportunity* requires only one of those two equalities to hold, whichever is considered more ethically important (e.g., in law enforcement, we may ensure that innocent individuals have similar treatment). Other notions of fairness include *predictive parity* [16], which matches classifier precision across groups, or the minimization of *mutual information* [34] between predictions and sensitive attributes. Except in degenerate cases, some pairs of criteria cannot be achieved simultaneously [3], meaning a domain expert must make a moral decision about which fairness constraint they aim to satisfy.

We formalize the problem of fair classification as learning the highest-accuracy model that satisfies a constraint on some fairness criterion (either demographic parity or equality of opportunity). This constrained optimization problem is mathematically difficult to solve, which has lead to the creation of a variety of fair classification algorithms, many involving *relaxations* of constraints like those analyzed in this paper. Prior work has mostly limited experiments to small tabular data sets like COMPAS [2], obscuring the fact that many previous methods are either heuristic, or have guarantees that are limited to linear classifiers with convex losses.

---

[*]Department of Computer Science, University of California Irvine School of Information and Computer Science, Irvine, CA, USA

35th Conference on Neural Information Processing Systems (NeurIPS 2021).

We compare fairness relaxations via an optimization framework that explores entire trajectories of fairness-accuracy trade-offs, illuminating advantages and weaknesses of each method. We are the first to show (theoretically and empirically) what conditions may lead relaxation methods to fail or converge to degenerate solutions. We further show that this problem grows worse for non-convex classifiers like deep neural networks. To address this weakness, we promote three new fairness surrogates with promising theoretical properties: the *log-sigmoid difference*, *log-sigmoid sum*, and *sigmoid* relaxations. These surrogates are more stable and efficient to optimize than prior work. The log-sigmoid sum and sigmoid bounds have provable fairness guarantees; the log-sigmoid difference has weaker guarantees, but excellent empirical performance.

We are the first to empirically compare relaxation methods on deep neural network models for large-scale image and text classification. To enable these experiments, we show how our fairness surrogates may be optimized via stochastic gradient descent, and applied to data where the sensitive attribute is not always observed. Our methods are comparable to the state-of-the-art on small tabular datasets, and show clear improvements for deep learning of flexible (but fair) image and text classifiers.

## 2 Background: Fair Classification

Let $\mathcal{D} = \{(\mathbf{x}, s, y)\}^N$ be a set of $N$ training examples with predictive features $\mathbf{x} \in \mathbb{R}^p$, target class $y \in \{0, 1\}$, and sensitive attribute $s \in \{a, b\}$ encoding protected group membership. A parameterized classifier $f_\theta(\mathbf{x})$, like logistic regression or a neural network, then predicts $y = 1$ when $f_\theta(\mathbf{x}) > 0$.

In this work, we focus on the fairness metrics *demographic parity* (DP) and *equality of opportunity* (EO). We define the *difference in demographic parity* (DDP) to equal

$$\text{DDP}(\theta) = P(f_\theta(X) > 0 \mid S = a) - P(f_\theta(X) > 0 \mid S = b). \tag{1}$$

If $\text{DDP}(\theta) \gg 0$, group $a$ receives an unfair share of positive outcomes; if $\text{DDP}(\theta) \ll 0$, group $b$ is favored. We seek models where $\text{DDP}(\theta) \approx 0$. The *difference in equality of opportunity* (DEO) is

$$\text{DEO}(\theta) = P(f_\theta(X) > 0 \mid S = a, Y = 0) - P(f_\theta(X) > 0 \mid S = b, Y = 0), \tag{2}$$

where fair classifiers have $\text{DEO}(\theta) \approx 0$. If false negative rates are of more concern than false positives, $\text{DEO}(\theta)$ may be modified to condition on $Y = 1$ instead of $Y = 0$. We seek the best classifier whose empirical fairness disparity $\Delta(\theta)$ has magnitude smaller than some tolerance $\epsilon > 0$:

$$\min_\theta \quad \frac{1}{N} \sum_{(\mathbf{x}, s, y) \in \mathcal{D}} \mathcal{L}(f_\theta(\mathbf{x}), y), \text{ such that } |\Delta(\theta)| \le \epsilon. \tag{3}$$

Our experiments and theorems focus on the case where $f_\theta(\mathbf{x}) = \mathbf{w}^T \Phi(\mathbf{x})$ with (possibly learned) data representation $\Phi(\mathbf{x})$, output layer weights $\mathbf{w}$, and overall parameters $\theta = \mathbf{w} \cup \theta_\Phi$. We let $\mathcal{L}(.)$ be the negative log-likelihood (or cross-entropy) loss,

$$\mathcal{L}(f_\theta(\mathbf{x}), y) = -y \log(\sigma(f_\theta(\mathbf{x}))) - (1 - y) \log(\sigma(-f_\theta(\mathbf{x}))),$$

with $\sigma(z) = 1/(1 + e^{-z})$. This model family includes logistic regression and many neural networks. We set $\Delta(\theta) = \widehat{\text{DDP}}(\theta)$ for demographic parity, $\Delta(\theta) = \widehat{\text{DEO}}(\theta)$ for equality of opportunity, where

$$\widehat{\text{DDP}}(\theta) = \frac{1}{N_{a.}} \sum_{\substack{(\mathbf{x}, s, y) \in \mathcal{D} \\ s = a}} 1_{\{f_\theta(\mathbf{x}) > 0\}} - \frac{1}{N_{b.}} \sum_{\substack{(\mathbf{x}, s, y) \in \mathcal{D} \\ s = b}} 1_{\{f_\theta(\mathbf{x}) > 0\}}, \tag{4}$$

$$\widehat{\text{DEO}}(\theta) = \frac{1}{N_{a0}} \sum_{\substack{(\mathbf{x}, s, y) \in \mathcal{D} \\ s = a, y = 0}} 1_{\{f_\theta(\mathbf{x}) > 0\}} - \frac{1}{N_{b0}} \sum_{\substack{(\mathbf{x}, s, y) \in \mathcal{D} \\ s = b, y = 0}} 1_{\{f_\theta(\mathbf{x}) > 0\}}. \tag{5}$$

There are $N_{sy}$ data with sensitive group $s$ and predictive label $y$, $N_{s.}$ data with group $s$ and any label, and $N_{.y}$ data with label $y$ and any group. For later use, we also define $p_{sy} = \frac{N_{sy}}{N}, p_{s.} = \frac{N_{s.}}{N}$, and $p_{.y} = \frac{N_{.y}}{N}$ as the proportions of data with each group and label. To achieve equalized odds, we may enforce two constraints as in Eq. (5), one for false positives and one for false negatives.

In addition to selecting the fairness metric, a user must specify an unfairness tolerance $\epsilon > 0$. Although we might wish to bring unfairness to exactly zero, this may not be possible except in trivial cases [22, 49], or might come at a huge accuracy cost [27] with no guarantee of generalized fairness on test data [32]. Moreover, positive tolerances often have legal justifications: adverse impact hiring laws dictate that the ratio of hiring rates for two sensitive groups should be at least 4/5 [21].

## 2.1 Surrogates for Tractable Optimization

The constrained optimization problem of Eq. (3) is typically intractable due to the discontinuous constraints $\Delta(\theta)$, which make gradient-based algorithms for constrained optimization inapplicable or ineffective. We overcome this intractability by replacing constraints with smooth surrogates.

**Linear relaxations.** The widely-used family of linear relaxations [52] seeks to improve demographic parity by replacing the intractable difference in acceptance rates with the covariance between the sensitive attribute and model output. Letting $z = 1$ if $s = a$ and $z = 0$ if $s = b$, we then have

$$\text{Cov}(z, f_\theta(\mathbf{x})) \approx \frac{1}{N} \sum_{i=1}^{N} (z_i - \bar{z}) f_\theta(\mathbf{x}_i) \propto \mathbb{E}[f_\theta(\mathbf{x})|Z = 1] - \mathbb{E}[f_\theta(\mathbf{x})|Z = 0]. \quad (6)$$

Because the sensitive attribute is binary, this covariance equals a difference in expectations. Bringing this *first moment relaxation* [49] to zero is equivalent to forcing the output distributions for the two sensitive groups to have equal means. By modifying the expectations in Eq. (6) to also condition on $Y = 0$, others have applied equivalent relaxations (up to constants) to equality of opportunity [4, 17]. However, matching the mean response across groups is insufficient to ensure fairness: Classifiers often still differ in higher moments, causing disparities on how groups are treated [40].

**Non-linear relaxations.** Zafar et al. [53, 51] also propose a relaxation for equality of opportunity, the covariance between the sensitive attribute and the model response *for only misclassified points*:

$$\text{Cov}(z, g(f_\theta(\mathbf{x})) \mid Y = 0) \propto \mathbb{E}[g(f_\theta(\mathbf{x})) \mid Z = 1, Y = 0] - \mathbb{E}[g(f_\theta(\mathbf{x})) \mid Z = 0, Y = 0], \quad (7)$$

where $g(f_\theta(\mathbf{x})) = \max(f_\theta(\mathbf{x}), 0)$. This constraint is non-convex, but for classifiers like SVMs with convex loss functions, convex-concave programming [45] may be used to find a local optimum. However, the gap between the relaxation and the true target persists: even when this covariance equals 0, we have no guarantees about the difference in false positive rates between the two groups.

## 2.2 Other Related Work

**Post-Processing of Unfair Classifiers.** Post-processing methods learn a model via an unconstrained loss with no regard to fairness, and then enforce fairness as a post-processing step [27] by setting group-specific decision boundaries to equalize false positive rates (or acceptance rates). However, this requires the sensitive attribute at test time. Besides being impractical in some applications, such classifiers may make different decisions for individuals with identical features $\mathbf{x}$. Such *disparate treatment* constitutes a different form of unfairness and can be illegal in some domains [19].

Wei et al. [48] avoid requiring sensitive attributes by training a separate classifier to estimate $p(S \mid X)$. Morally, this is not much different than disparate treatment; instead of treating individuals differently based on their sensitive attributes, the model treats them differently based on its estimate of their sensitive attribute. The authors also acknowledge that while their method is effective at small reductions in unfairness, it often fails to achieve large reductions. Without knowledge of true sensitive attributes, post-processing algorithms may lack the maneuverability to dramatically reduce unfairness.

**Fairness by Density Estimation.** Celis et al. [9] optimize classifiers with a constraint on the true fairness discrepancy of the data-generating distribution. This requires an explicit density estimate for the training data, for which they use a Gaussian Naive Bayes model that would poorly capture high-dimensional images or text. Generative modeling of such data is a challenging research problem.

**Data Set Stability.** Huang & Vishnoi [31] examine the stability of fair classifiers under data perturbations, and propose a regularizer that can be added to other fairness algorithms, including ours. We instead analyze and improve fairness surrogates for complex, non-convex classifiers.

**Fair Mixup.** Instead of penalizing a surrogate to the fairness discrepancy, Chuang & Mroueh [11] penalize the gradient of $f_\theta(X)$ along interpolated paths between data with opposite attributes. This approach is less interpretable, and does not guarantee demographic parity nor equality of opportunity.

**Fairness in Computer Vision**. Large-scale image data sets are often unannotated and thus lack guarantees that they are balanced along sensitive attributes such as race or gender [10]. Dataset balance has become a focus of conversation in fair computer vision [29], but it is insufficient to guarantee fairness [47]. Other work in fair computer vision has explored issues of bias amplification [28, 55], counterfactual fairness [14], and transfer learning [18, 35]. Whereas relaxation methods explore fairness-accuracy trade-offs, many vision papers use fairness as a vehicle to boost accuracy [44, 38,

46]. By improving classifier performance on under-represented sub-populations, overall accuracy can be increased, but groups may still receive disparate impact from the model's decisions.

**Fairness in Natural Language Processing.** Much of the previous work on fair language models has focused on domain-specific examples of bias, such as word embeddings [5, 57, 25, 6, 58] or coreference resolution [43, 56]. In contrast, we are not concerned with latent bias in the language model's word representation; we want to ensure that predictive models do not discriminate against authors based on their identity. It has been shown that gender expression can be predicted from writing [37], so models could discriminate against authors of one gender if fairness is not mandated.

## 3  A Lagrangian Framework for Relaxations

The absolute value in Eq. (3) induces two linear fairness constraints, a lower bound and an upper bound, at most one of which can be active. For the optimal unconstrained classifier parameters $\theta^* = \arg\min \frac{1}{N} \sum_{(\mathbf{x},s,y)\in\mathcal{D}} \mathcal{L}(f_\theta(\mathbf{x}), y)$, we assume (without loss of generality) that $\Delta(\theta^*) > \epsilon$. If $\Delta(\theta^*) < -\epsilon$, we swap the sensitive groups $a, b$ to negate the discrepancy and obtain $\Delta(\theta^*) > \epsilon$. If $|\Delta(\theta^*)| \leq \epsilon$, the unconstrained classifier is already sufficiently fair and optimally solves Eq. (3).

### 3.1  Fair Classification via Smooth Penalties

Letting $g(r)$ be a continuous and monotonically non-decreasing surrogate for the indicator functions arising in exact fairness constraints, we define the following tractable optimization problem:

$$\min_\theta \ \frac{1}{N} \sum_{(\mathbf{x},s,y)\in\mathcal{D}} \mathcal{L}(f_\theta(\mathbf{x}), y), \ \text{s.t. } \Delta_g(\theta) \leq \tau. \tag{8}$$

For demographic parity, we set $\Delta_g(\theta) = \widehat{\text{DDP}}_g(\theta)$, where

$$\widehat{\text{DDP}}_g(\theta) = \frac{1}{N_{a.}} \sum_{\substack{(\mathbf{x},s,y)\in\mathcal{D} \\ s=a}} g(f_\theta(\mathbf{x})) - \frac{1}{N_{b.}} \sum_{\substack{(\mathbf{x},s,y)\in\mathcal{D} \\ s=b}} g(f_\theta(\mathbf{x})). \tag{9}$$

For equality of opportunity, we similarly define $\widehat{\text{DEO}}_g(\theta)$ by replacing the indicators in Eq. (5) with surrogates $g(r)$. When $g(r) = r$, for DDP we recover [53], and for DEO we recover [4, 17]. In this case, both $g(r)$ and $-g(r)$ are linear and convex, so when the loss is also convex we have a convex optimization problem. When $g(r) = \max(0, r)$, for DEO we recover the relaxation in [51, 53]. For non-linear $g(r)$, Eq. (8) is non-convex even for convex losses due to the $-g(r)$ term in $\Delta_g(\theta)$.

Unlike some prior work, we employ the KKT conditions [42] to optimize Eq. (8) via its Lagrangian:

$$\min_\theta \ \ \mathcal{L}_{g,\lambda}(\theta), \ \ \text{where } \mathcal{L}_{g,\lambda}(\theta) = \frac{1}{N} \sum_{(\mathbf{x},s,y)\in\mathcal{D}} \mathcal{L}(f_\theta(\mathbf{x}), y) + \lambda\Delta_g(\theta). \tag{10}$$

For every relaxed-fairness tolerance $\tau$, there exists a $\lambda \geq 0$ such that the constrained optimum of (8) is a minimizer of (10). Furthermore, second-order sufficient conditions [42, Chap. 12.5] guarantee that every local minima $\hat{\theta}$ of (10), with fairness discrepancy $\Delta_g(\hat{\theta})$, is a feasible local minima of the constrained objective (8) with $\tau = \Delta_g(\hat{\theta})$. For non-convex losses $\mathcal{L}$, gradient descent may reach a fixed point with sub-optimal likelihood. But this is a concern for any optimizer, and in experiments we find that deep neural networks are flexible enough to achieve effective fairness-accuracy tradeoffs.

We can rewrite the fairness-penalized objective of Eq. (10) as a sum over data examples:

$$\mathcal{L}_{g,\lambda}(\theta) = \frac{1}{N} \sum_{(\mathbf{x},s,y)\in\mathcal{D}} \left( \mathcal{L}(f_\theta(\mathbf{x}), y) + \lambda R_g(\mathbf{x}, y, s) \right). \tag{11}$$

For demographic parity and equality of opportunity, the fairness regularizers are given by

$$R_g(\mathbf{x}, y, s) = \begin{cases} \frac{g(f_\theta(\mathbf{x}))}{p_{a.}} & \text{if } s = a, \\ -\frac{g(f_\theta(\mathbf{x}))}{p_{b.}} & \text{if } s = b. \end{cases} \ (DP), \quad R_g(\mathbf{x}, y, s) = \begin{cases} \frac{g(f_\theta(\mathbf{x}))}{p_{a0}} & \text{if } s = a, y = 0, \\ -\frac{g(f_\theta(\mathbf{x}))}{p_{b0}} & \text{if } s = b, y = 0, \ (EO). \\ 0 & \text{otherwise.} \end{cases}$$

Our framework differs from prior work that introduced these surrogates in that we do not seek $\Delta_g = 0$. We know $\Delta_g$ will decrease as we increase $\lambda$, but it might be necessary to bring $\Delta_g < 0$ to achieve $\Delta \leq \epsilon$ for relaxations (like the linear) where $\Delta_g$ does not rigorously bound $\Delta$. In practice, we need not monitor $\Delta_g$: we can increase $\lambda$ and track changes in $\Delta$ until we find a $\lambda^*$ with $\Delta \leq \epsilon$, if it exists.

## 3.2 Degeneracy of Fairness Relaxations

When we choose an unbounded surrogate, we may observe pathological behavior where $\Delta_g$ becomes negative and large enough in magnitude for the optimizer to ignore predictive loss. For example in Fig. 1, the linear relaxation performs reasonably well on an initial synthetic dataset. We then add an outlier with $(x_1, x_2) = (10^3, 10^3)$ and $y = 1$. The unconstrained model sees no change, because the new point is already classified correctly. If the outlier has $s = b$, then the relaxed difference becomes much more negative. This outlier causes the classifier to degenerate for even small $\lambda$ values, focusing on improving the fairness objective at great detriment to the predictive objective.

The optimization framework of Sec. 3.1 may fail in two ways: the surrogate problem (8) could be infeasible if the model is not expressive enough, or a poor choice of surrogate $g(r)$ may cause there to be no value of $\tau$ for which the solution to (8) solves the true objective (3). We now provide theoretical conditions under which fairness surrogates fail to achieve fairness. Detailed proofs are in the supplement.

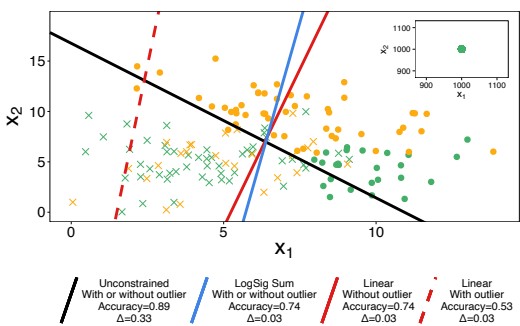

Figure 1: Synthetic dataset with large latent fairness discrepancy. Circles have true label $y = 1$ and crosses have true label $y = 0$. Color encodes sensitive attribute, with green indicating group $b$. For both linear and log-sigmoid sum (see Sec. 4.2) fairness relaxations, we trained a logistic regression model, searching across a grid of $\lambda$ values to find one for which $\widehat{\text{DDP}} \leq 0.03$. Adding a single outlier at $(10^3, 10^3)$ with $y = 1, s = b$ does not change the unconstrained or log-sigmoid decision boundaries, but linear deteriorates dramatically.

We minimize the Lagrangian $\mathcal{L}_{g,\lambda}$ of Eq. (11) with predictors $\Phi(\mathbf{x})$ and a fairness surrogate $g(r)$ that is continuous and monotonically increasing, where limits $\delta_g^+ = \lim_{r \to \infty} g'(r)$ and $\delta_g^- = \lim_{r \to -\infty} g'(r)$ exist. Let $\mu_{jsy}$ be the mean of $\Phi(\mathbf{x})_j$, the $j^{\text{th}}$ entry of $\Phi(\mathbf{x})$, for the $N_{sy}$ data points with sensitive group $s$ and label $y$. Also let $\mu_{js\cdot}$ be the mean of $\Phi(\mathbf{x})_j$ for data with group $s$ and any label, and $\mu_{j\cdot y}$ be the mean of $\Phi(\mathbf{x})_j$ for data with label $y$ and any group.

**Theorem 1** (Degeneracy for DP). *Consider a feature $j$ such that $\Phi(\mathbf{x})_j \geq 0$ for all $\mathbf{x} \in \mathcal{D}$.*
*(1) If $\delta_g^- \neq 0$ and $\mu_{ja\cdot} > \mu_{jb\cdot}$, then for $\lambda > \lambda_j^* = \frac{\mu_{j\cdot 1} p_{\cdot 1}}{\delta_g^-(\mu_{ja\cdot} - \mu_{jb\cdot})}$, $\lim_{\mathbf{w}_j \to -\infty} \mathcal{L}_{g,\lambda}(\theta) = -\infty$.*
*(2) If $\delta_g^+ \neq 0$ and $\mu_{ja\cdot} < \mu_{jb\cdot}$, then for $\lambda > \lambda_j^* = \frac{\mu_{j\cdot 0} p_{\cdot 0}}{\delta_g^+(\mu_{jb\cdot} - \mu_{ja\cdot})}$, $\lim_{\mathbf{w}_j \to +\infty} \mathcal{L}_{g,\lambda}(\theta) = -\infty$.*

**Theorem 2** (Degeneracy for EO). *Consider a feature $j$ such that $\Phi(\mathbf{x})_j \geq 0$ for all $\mathbf{x} \in \mathcal{D}$.*
*(1) If $\delta_g^- \neq 0$ and $\mu_{ja0} > \mu_{jb0}$, then for $\lambda > \lambda_j^* = \frac{\mu_{j\cdot 1} p_{\cdot 1}}{\delta_g^-(\mu_{ja0} - \mu_{jb0})}$, $\lim_{\mathbf{w}_j \to -\infty} \mathcal{L}_{g,\lambda}(\theta) = -\infty$.*
*(2) If $\delta_g^+ \neq 0$ and $\mu_{ja0} < \mu_{jb0}$, then for $\lambda > \lambda_j^* = \frac{\mu_{j\cdot 0} p_{\cdot 0}}{\delta_g^+(\mu_{jb0} - \mu_{ja0})}$, $\lim_{\mathbf{w}_j \to +\infty} \mathcal{L}_{g,\lambda}(\theta) = -\infty$.*

For constraint multipliers $\lambda > \lambda_j^*$, the Lagrangian is unbounded below and can be reduced arbitrarily by changing a single parameter $\mathbf{w}_j$. Each attribute $j$ satisfying (1) or (2) induces a critical $\lambda_j^*$, so the entire model has a critical value $\lambda^* = \min_j \lambda_j^*$, where we take $\lambda_j^* = \infty$ for an attribute $j$ not satisfying either criterion. Because the predictive loss is bounded below, divergence must be caused by $\Delta_g$ approaching $-\infty$. Note this divergence is *not* a failure of the KKT conditions: for every finite $\tau$, the corresponding constraint multiplier $\lambda$ is bounded in $0 \leq \lambda \leq \lambda^*$. Further note that if $\Phi(\mathbf{x})$ is fixed and $g(r)$ is linear, $\mathcal{L}_{g,\lambda}$ is convex and has no stationary points for all $\lambda > \lambda^*$.

While we may solve the surrogate problem (11) for any finite $\tau$ via some $\lambda \leq \lambda^*$, for general $g(r)$ there is no guarantee we satisfy our true fairness constraints (3). This is illustrated in Fig. 2, which highlights gaps between the surrogate and true rate differences. Degeneracy occurs for the linear relaxation $g(r) = r$, where $\delta_g^- = \delta_g^+ = 1$, of DDP on the Adult data set. For the ReLU relaxation of DEO, which takes $g(r) = \max(0, r)$ so that $\delta_g^- = 0$ but $\delta_g^+ = 1$, degeneracy also occurs on the COMPAS data set; this is possible because $-g(r)$ and $\Delta_g$ are unbounded below. Although our theorem provides a sufficient but not necessary condition for degeneracy, in Fig. 2 it accurately predicts the value of $\lambda$ for which degeneracy occurs, suggesting the bound may sometimes be tight.

Even in less extreme cases like Fig. 1 where we recover a solution to the true target, if we need to set $\tau \ll 0$ and thus $\lambda \approx \lambda^*$, the surrogate's magnitude may cause the learned model to mostly ignore predictive loss. While $L_2$ regularization will prevent $\mathbf{w}_j$ from fully diverging, it may still

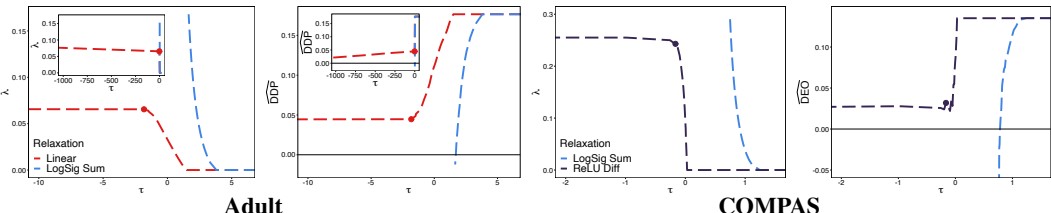

**Figure 2:** Relationship between surrogate threshold $\tau$ and multiplier $\lambda$ (first & third plots) or non-relaxed unfairness (second & fourth plots). When $\tau$ is large, the constraint is inactive. Highlighted points show models with critical $\lambda = \lambda^*$ derived from Theorem 1 (left, for DDP) or Theorem 2 (right, for DEO). Our log-sigmoid sum surrogate provably never degenerates (see Sec. 4.2). We avoid numerical instabilities via very weak $L_2$-regularization of weights $\mathbf{w}$, with coefficient $2^{-20}$, so that models with $\lambda > \lambda^*$ have bounded loss.

grow large enough in magnitude to substantially hurt performance. For example, when constraining DDP on the Adult dataset, the feature `occupation_Protective-serv` induces $\lambda^* \approx 0.066$. Regularizing above that threshold results in huge losses to mean log-likelihood as $\mathbf{w}_j$ diverges (see Fig. 3). This degeneracy is not clearly visible from accuracy plots because only 4% of the data has `occupation_Protective-serv=1`.

Degeneracy becomes more prevalent for deep neural networks, which can learn one feature $\Phi(\mathbf{x})_j$ such that $\lambda^*$ is minimized and very negative training loss is achievable, while doing little to improve the true fairness. The assumption that $\Phi(\mathbf{x})_j \geq 0$ is satisfied by many common attributes including discrete indicators and ReLU non-linearities in neural networks. As we prove in the supplement, notationally complicated generalizations of Theorems 1-2 exist that characterize arbitrary $\Phi(\mathbf{x})$.

### 3.3 Related Work: SearchFair

The SearchFair algorithm [40] also reduced relaxations $\Delta_g < 0$ to enforce fairness. Letting $\theta(\lambda)$ be the model parameters induced by constraint multiplier $\lambda$, they prove that when the fairness-penalized loss is *convex*, $\mathrm{DDP}(\theta(\lambda))$ for the (unknown) true data distribution is continuous with respect to $\lambda$. If there exist $\lambda^-$ and $\lambda^+$ such that $\mathrm{DDP}(\theta(\lambda^-)) < 0$, $\mathrm{DDP}(\theta(\lambda^+)) > 0$, there exists a root $\lambda^- < \lambda^\circ < \lambda^+$ with $\mathrm{DDP}(\theta(\lambda^\circ)) = 0$. SearchFair then uses binary search to approximate $\lambda^\circ$.

A limiting assumption of SearchFair is that $\lambda^-$ and $\lambda^+$ exist for any fairness surrogate. One of our key contributions is to demonstrate theoretically (via Theorems 1-2) and empirically (via Figs. 2-3) that this is not true: binary search cannot be used to find the root of a function with no root. Note that Lohaus et al. [40] explicitly consider both relaxations whose degeneracies are demonstrated in Fig. 2.

Continuity of $\mathrm{DDP}(\theta(\lambda))$, as required to guarantee the existence of the $\lambda^\circ$ sought by SearchFair, is *not* guaranteed for non-convex losses like (deep) neural networks. We do not use SearchFair's binary search for this reason, and because (as motivated in Sec. 2) we instead seek $|\mathrm{DDP}(\theta)| \leq \epsilon$ for $\epsilon > 0$.

## 4 Logistic Fairness Relaxations

We promote surrogates based on the *log-sigmoid* $g(r) = -\log\sigma(-r)$ and *sigmoid* $g(r) = \sigma(r)$.

### 4.1 Log-Sigmoid Differences for Improved Fairness

**Difference in Equality of Opportunity.** The log-sigmoid surrogate $g(r) = -\log\sigma(-r)$ has the same form as the log-likelihood: $g(f_\theta(\mathbf{x})) = \mathcal{L}(f_\theta(\mathbf{x}), 0)$. When using that loss function and the DEO criterion, the two terms in Eq. (11) combine nicely, and the relaxed objective can be interpreted as a simple reweighting of points: positive examples have weight 1, negative examples of group $a$ have weight $1 + \frac{\lambda}{p_{a0}}$, and negative examples of group $b$ have weight $1 - \frac{\lambda}{p_{b0}}$. Thus for logistic regression with $\lambda \leq p_{b0}$, the Lagrangian formulation in Eq. (10) is *convex* and bounded even though the corresponding constraints in Eq. (8) are non-convex. This analysis aligns with our earlier theorems:

**Theorem 3.** *For any surrogate $g(r)$ of EO such that $\delta_g^- = 0$, any feature $\Phi(\boldsymbol{x})_j$ satisfying the conditions of Thm. 2 has a bounded degeneracy threshold $\lambda_j^* \geq \frac{p_{b0}}{\delta_g^+}$.*

The reweighting interpretation shows that the log-sigmoid surrogate, which has $\delta_g^- = 0, \delta_g^+ = 1$, cannot degenerate for $\lambda < p_{b0}$. Our theorem recovers that result, and extends it to other surrogates with the same asymptotic behavior (such as a ReLU function). Note that for $\lambda > \lambda^*$, degeneracy as in Fig. 2 is possible as we observe for the ReLU surrogate, but this is not observed for log-sigmoid in any of our experiments. Surrogates with $\delta_g^- \neq 0$ observe no such lower bound, and can degenerate for any $\lambda > 0$. Because a neural network can learn a representation to minimize $\lambda^*$ and thus diverge

the objective, Theorem 3's guarantee that *any* learned representation $\Phi(\mathbf{x})$ exhibits a threshold above $p_{b0}$ is powerful. A generalization of Thm. 3, that allows arbitrary $\Phi(\mathbf{x})$, is proven in the supplement.

**Difference in Demographic Parity.** To retain this intuitive reweighting property for DP, we need to keep $g(r)$ in the same form as the loss, which depends on the true label $y$. Thus, we must use a different surrogate for each target label. Suppose we use the surrogate $g(r) = -\log \sigma(-r)$ for data with $y = 0$, and $g(r) = \log \sigma(r)$ for data with $y = 1$. Because the surrogates are of the same form as the predictive loss, we recover an unconstrained classification problem where negative examples of group $a$ have weight $1 + \frac{\lambda}{p_{a.}}$, negative examples of group $b$ have weight $1 - \frac{\lambda}{p_{b.}}$, positive examples of group $a$ have weight $1 - \frac{\lambda}{p_{a.}}$ and positive examples of group $b$ have weight $1 + \frac{\lambda}{p_{b.}}$. Thm. 1 assumes a single surrogate and does not apply here, but an expanded theorem is included in the supplement. We note that degeneracy is not possible for $\lambda \le \min\{p_{b.}, p_{a.}\}$ because the reweighting intepretation gives us all non-negative weights, and therefore a convex and bounded loss.

**Related Work: Data Re-weighting.** Re-weighting of data points is a classic way to correct deficiencies of generative models [30, 26]. For fair classification, Kamiran & Calders [33] pre-process data to encourage demographic parity by assigning weight $w(s, y) = \frac{p(s)p(y)}{p(s \wedge y)}$ to points from group $s$ with label $y$. Probabilities are estimated from the training data so that $y$ and $s$ are independent under the weighted empirical distribution. However, this lack of dependence does not guarantee demographic parity; relative to these weights, classifiers may either under- or over-balance sensitive groups.

Our log-sigmoid difference relaxation instead implies that *some* amount of weight increase for the low-accuracy group, and weight decrease for the high-accuracy group, leads to approximate fairness. The search for $\lambda$ must be integrated with training; it cannot be determined from raw data statistics.

**Related Work: Reductions Approach.** Agarwal et al. [1] use the 0-1 loss for both the predictive loss and the constraint. These losses linearly combine via the KKT conditions, inducing a classification problem with reweighted data. They then relax the combined loss via either a hinge or logistic log-likelihood surrogate. Their Fair Reductions approach is thus closely related to our log-sigmoid difference surrogate, but the two methods are not exactly equivalent: their fairness criteria are defined via distinct constraints, and for constraint multipliers that induce negative weights for some training examples, they instead swap the example's label $y$ and assign positive weights.

## 4.2 Log-Sigmoid Sums with Fairness Guarantees

The primary weakness of Eq. (8) is that we lack meaningful guarantees about the relationship between $\Delta_g$ and $\Delta$. When the fairness criterion is DP, we can address this problem by rewriting the unfairness

$$\widehat{\mathrm{DDP}}(\theta) = \sum_{\substack{(\mathbf{x},s,y) \in \mathcal{D} \\ s=a}} \frac{\mathbf{1}_{\{f_\theta(\mathbf{x})>0\}}}{N_{a.}} + \sum_{\substack{(\mathbf{x},s,y) \in \mathcal{D} \\ s=b}} \frac{\mathbf{1}_{\{-f_\theta(\mathbf{x})>0\}}}{N_{b.}} - 1. \tag{12}$$

Replacing indicators with convex upper bounds $g(r)$, the corresponding Lagrangian becomes:

$$\min_{\theta} \quad \frac{1}{N} \sum_{(\mathbf{x},s,y) \in \mathcal{D}} \mathcal{L}(f_\theta(\mathbf{x}), y) \text{ such that } \sum_{\substack{(\mathbf{x},s,y) \in \mathcal{D} \\ s=a}} \frac{g(f_\theta(\mathbf{x}))}{N_{a.}} + \sum_{\substack{(\mathbf{x},s,y) \in \mathcal{D} \\ s=b}} \frac{g(-f_\theta(\mathbf{x}))}{N_{b.}} \le \tau + 1. \tag{13}$$

We need $g(r) \ge \mathbf{1}_{\{r>0\}}$ for all $r \in \mathbb{R}$, as satisfied by the hinge function $g(r) = \max(0, 1 + r)$ and the scaled log-sigmoid $g(r) = -c \log \sigma(-r)$, $c = 1/\log(2)$. Because the constraint in Eq. (13) is a rigorous upper bound on $\widehat{\mathrm{DDP}}(\theta)$, if we set $\tau \le \epsilon$, we are guaranteed that $\widehat{\mathrm{DDP}}(\theta) \le \Delta_g(\theta) \le \tau \le \epsilon$. This bound prevents the pathological behavior we demonstrated for other relaxations, where $\Delta_g$ can be brought arbitrarily negative without sufficiently reducing $\widehat{\mathrm{DDP}}$. An identical derivation can be applied to the EO criterion, resulting in a surrogate involving only the points with $y = 0$.

In practice, the Lagrangian lets us skip the need to select $\tau$ entirely. For expressive models where (13) is feasible, there exists some surrogate threshold $\tau$, and therefore $\lambda$, which reduces fairness discrepancy below $\epsilon$. A practitioner must simply search $\lambda$ to find a value satisfying their tolerance $\epsilon$.

We can rewrite Eq. (13) as in Eq. (11), but with a new fairness regularizer $R_g^+$:

$$R_g^+(\mathbf{x}, y, s) = \begin{cases} \frac{g(f_\theta(\mathbf{x}))}{p_{a.}} & \text{if } s = a, \\ \frac{g(-f_\theta(\mathbf{x}))}{p_{b.}} & \text{if } s = b. \end{cases} (DP), \quad R_g^+(\mathbf{x}, y, s) = \begin{cases} \frac{g(f_\theta(\mathbf{x}))}{p_{a0}} & \text{if } s = a, y = 0, \\ \frac{g(-f_\theta(\mathbf{x}))}{p_{b0}} & \text{if } s = b, y = 0, \quad (EO). \\ 0 & \text{otherwise.} \end{cases}$$

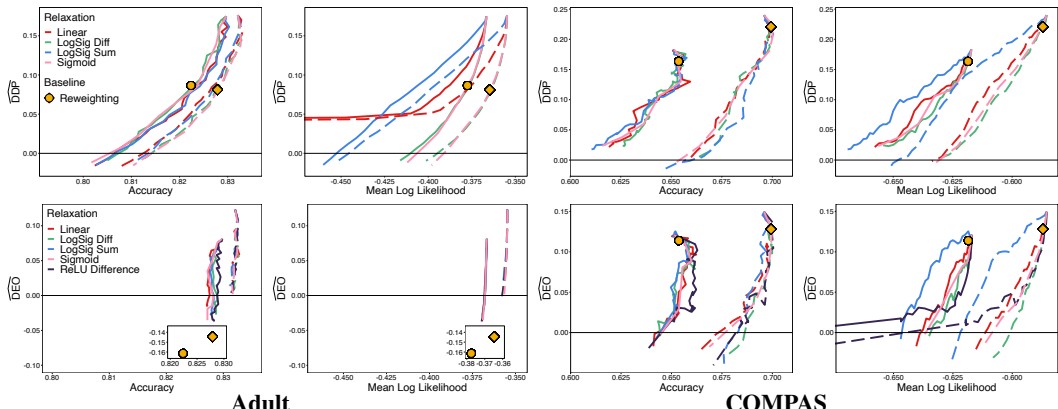

Figure 3: Fair logistic regression results on the Adult and COMPAS data sets. Each relaxation is run over a dense grid of $\lambda$ values until $\Delta(\theta) < 0$ is achieved on training data (or for demographic parity on Adult, until degeneracy occurs as in Fig. 2). Dashed lines show results on training data, solid lines on test data. For the reweighting baseline [33], the diamond shows training results and the circle test results.

**Related Work: Lagrangian Dual Ascent.** The Lagrangian dual [1, 41] provides an iterative approach to choosing Lagrange multipliers for multiple constraints. A grid search over $\lambda$ values is usually sufficient for a single constraint [1], but scales exponentially with the number of constraints. Some prior work has focused on efficiently setting $\lambda$ [1, 12, 40, 41], but all of these methods require the use of a surrogate function, and none of them discuss the desiderata for that choice. Thus, we view our work as orthogonal to theirs; surrogates informed by our theory may be integrated with a more complex dual ascent method when multiple fairness constraints are enforced.

**Related Work: Lower-Upper Bounds.** Wu et al. [50] used a similar transform to replace the difference in rates with a sum. They then add a second constraint to Eq. (13), using a concave surrogate $h(r) \leq 1_{\{r>0\}}$ to lower bound $\widehat{DDP}(\theta)$, giving us $-\widehat{DDP}(\theta) \leq -\Delta_h(\theta) \leq \gamma$ for some lower-bound constraint threshold $\gamma$. Their experiments take $g(r) = \max(r + 1, 0)$ and $h(r) = -g(-r) + 1$. If this optimization problem is solved, we know the resulting model has $-\gamma \leq \widehat{DDP} \leq \tau$. However, choosing $\gamma$ and $\tau$ is difficult: hinge functions are loose bounds on the 0-1 loss, so typically $-\gamma \ll \widehat{DDP} \ll \tau$. Lohaus et al. [40] argue that for SVMs and meaningful values of $\gamma, \tau$, the constraints induce disjoint sets and no solution exists. Compared to this method, we identify the active fairness constraint and only require one upper bound. We also avoid the need to explicitly specify a threshold $\tau$ (let alone $\gamma$) by directly varying the constraint multiplier $\lambda$. This simplification is critical in enabling tractable application to more complex classifiers like deep neural networks.

### 4.3 Sigmoid Sums for Tight Bounds

We examine a third and final relaxation, the non-convex sigmoid function $g(r) = \sigma(r)$. Because $-\sigma(r) = \sigma(-r) - 1$, the difference and sum constructions lead to the same constraint. When viewed as a difference, we see $\delta_g^- = \delta_g^+ = 0$, preventing degeneracy. When viewed as a sum, we see it as a scaled upper bound on the indicator function, $2\sigma(r) \geq 1_{\{r>0\}}$. Thus like log-sigmoid sum, there provably exists some $\tau$ such that the solution to the surrogate achieves the true target fairness.

While sigmoids are widely used to relax indicator functions, there seems to be little work using them as surrogates for DDP or DEO. Kamishima et al. [34] use sigmoids to parameterize a different, mutual information fairness criterion. The cost of this approach is that $g(r)$ and $R_g(\mathbf{x}, y, s) = R_g^+(\mathbf{x}, y, s)$ are non-convex, producing local minima even for convex predictive losses. For deep neural networks, the vanishing gradients of the sigmoid function may lead to greater optimization difficulties.

## 5 Experiments and Data Sets

We compare relaxations by training fairness-regularized logistic regression on tabular data, and fairness-regularized deep neural networks on image and text data. To ensure equal treatment, for all models the sensitive attribute is not used as a predictive attribute. Methods for hyperparameter selection, data pre-processing, and optimization are detailed in the supplement.

**Relaxations.** We examine the *linear* relaxation $g(r) = r$ [52, 4, 17]; the non-linear *ReLU Diff* relaxation of [53, 51] with $g(r) = \max(0, r)$; our log-sigmoid difference (*LogSig Diff*) formulation

with $g(r) = -\log \sigma(-r)$ for DEO and $g(r|y=0) = -\log \sigma(-r), g(r|y=1) = \log \sigma(r)$ for DDP; our log-sigmoid sum (*LogSig Sum*) upper bound $g(r) = -\log \sigma(-r)$ with regularizer $R_g^+$; and our *sigmoid* upper bound $g(r) = \sigma(r)$. As a final baseline, we test the reweighting technique of Kamiran & Calders [33] as outlined in Sec. 4.1.

## 5.1 Fair Logistic Regression for Tabular Data

For fairness-regularized logistic regression we optimize Eq. (11) with negative log-likelihood loss function, linear model class $f_\theta(\mathbf{x}) = \theta^T \mathbf{x}$, and an added $L_2$ regularizer.

**Adult.** The Adult data set [36] is one of the most popular in the fair classification literature [1, 4, 12, 17, 23, 24, 40, 50, 53]. The goal of the prediction task is to predict whether an individual is earning more or less than $50,000 per year. We use sex as a sensitive attribute with values *male* and *female*.

**COMPAS.** The COMPAS data set was compiled by Angwin et al. [2] to investigate racial bias in recidivism prediction. The goal of the classification task is to predict whether the defendant will commit another crime or *recidivate* within two years. We use only the subset of the data with sensitive attribute *Caucasian* or *African-American*.

## 5.2 Fair Deep Learning with Image and Text Data

To our knowledge, we are the first to compare the performance of general-purpose fairness relaxations on deep neural networks for text and image classification. For the computer vision tasks, we use a randomly-initialized wide residual network architecture (WRN-50-2), which in 2016 broke established records on many visual recognition benchmarks including CIFAR-100 [54]. For the text classification task, we use Google's transformer-based BERT, a state-of-the-art language model [15].

**CelebA.** The CelebFaces Attributes data set [39] contains images of celebrities' faces with 40 annotated binary attributes. As in other work, we use *smiling* as the target attribute and *sex* as the sensitive attribute. We discard the remaining binary attributes and use the image data for prediction.

**Faces of the World.** The Faces of the World (FOTW) data set [20] comes from the 2016 Looking at People CVPR Challenge. We use *smiling* as the target attribute and *gender* as the sensitive attribute. To demonstrate the performance cost of Lohaus et al. [40]'s convexity requirement, we also compare our fairness-regularized deep neural network to their kernel-based approach.

**Yelp Text Data.** We use a subset of the Yelp review data set [13], which includes the text of reviews along with metadata including the reviewer's first name and the review's star rating. We perform the sentiment analysis task of predicting if a reviewer gives $\geq 4$ or $< 4$ stars from the text of the review. We use *gender* as the sensitive attribute, which is inferred from reviewers' first names.

**Sensitive Attribute Unknown.** The Yelp and FOTW data sets include examples for which the sensitive attribute is unknown or cannot be inferred. These data therefore do not contribute towards the fairness constraint, but do affect the predictive loss and improve classifier accuracy.

**Minibatch loss.** Because we have written the objective of Eq. (11) as a sum over data points, we can create an unbiased estimate of it from mini-batches $\mathcal{D}^{\{1\}}, \ldots, \mathcal{D}^{\{T\}}$, where $\mathcal{D}^{\{t\}} = \{(\mathbf{x}, s, y)\}^M$ has $M$ points drawn randomly from the full data set. By estimating via mini-batch the point-wise sum in Eq. (11), we are not evaluating the batch's surrogate discrepancy $\Delta_g$. Instead, by using the $N_{sy}$ and $p_{sy}$ statistics for the entire data set (not just the batch), we evaluate the batch's contribution to the full data unfairness $\Delta_g$. This is possible because demographic parity and equality of opportunity decompose as sums over data examples. This scalable stochastic approximation cannot be performed for other fairness criteria like predictive parity [16] or mutual information [34].

## 5.3 Analysis of Results

**Linear Degeneracy.** On CelebA, the linear relaxation degenerates almost immediately, achieving mean log-likelihoods 1000 times worse than competitors. This is predicted by our claim that neural networks may design $\Phi(\mathbf{x})$ to induce degeneracy as in Theorems 1-2. Degeneracy is less severe for FOTW and Yelp, which are initialized from pre-trained models. We conjecture that pre-training safeguards against the model learning to optimize fairness without regard for prediction.

**Smoothness Matters.** The log-sigmoid difference and ReLU difference surrogates have the same asymptotic limits, and therefore the same potential for degeneracy with large constraint multipliers. However, we only observe degeneracy for the ReLU difference formulation. Due to a mismatch between the smooth predictive loss and the ReLU surrogate, the Lagrangian may be non-convex

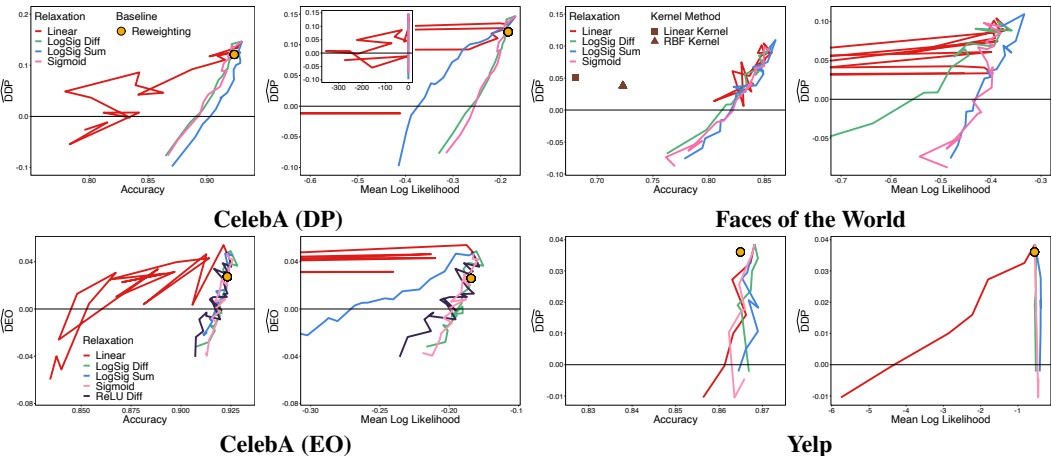

Figure 4: Fairness-regularized deep neural network results on computer vision and natural language processing data sets. Each simulation uses different random seeds for drawing minibatches, revealing instability in some relaxations. All results are shown on test data. The kernel-based model optimizes a convex loss function, using the SearchFair method of binary search to find a model with $\Delta(\theta) = 0$ on training data.

for any $\lambda > 0$. In the supplement, additional experiments show that optimization convergence for non-smooth surrogates is about 10 times slower than for smooth log-sigmoids.

**The Cost of Guarantees.** The log-sigmoid sum formulation is guaranteed to not degenerate, but this comes at the cost of poorer likelihood. We conjecture that this is due to the surrogate punishing high-confidence model outputs. The sigmoid formulation has the same fairness guarantees, but has slightly worse accuracy for large-scale data; we conjecture this is due to additional local optima.

**The Lagrangian View.** The re-weighting baseline often only reduces the fairness discrepancy marginally: data set balance is insufficient to ensure fairness. This baseline can be thought of as a way to set $\lambda$ before learning based on data statistics. Its poor fairness gains argue in favor of tuning multipliers via iterative training, despite the computational demands of training several models.

**The Cost of Convexity.** The binary search method of SearchFair [40] is only guaranteed to find a root when the loss function is convex. In the case of image data, forcing a linear model by using a kernel method comes at a large accuracy cost. Though SearchFair finds a point with $\Delta(\theta) = 0$ on the training split, due to sampling noise this does not guarantee perfect fairness on test data.

**Limitations.** Our strongest theoretical guarantees are about global minima, which non-convex optimization may fail to find. Training may converge to a model with arbitrarily worse likelihood than the true minimizer of Eq. (8). However, this is a problem for all deep learning methods. Because our surrogates induce losses which have been widely studied in deep learning, we conjecture that the same architectures that enable standard deep learning will be effective for our fair extensions.

We also acknowledge that fairness constraints are not a universal solution to problems of bias in artificial intelligence. Some tasks might be inherently unethical to automate, or the very existence of the task itself might be unethical and reinforce existing power structures. Demonstrating equitable group-level performance does not solve these problems, and might obfuscate deeper problems.

## 6 Discussion

We analyze three new fairness surrogates: the log-sigmoid difference, log-sigmoid sum, and sigmoid relaxations. They are easier to optimize than non-smooth alternatives in the literature, and provably more stable (especially for non-convex classifiers) than linear or ReLU relaxations. Experiments demonstrate the tightness of our theoretical results, advancing understanding of trade-offs between fairness surrogates. Results on complex vision and NLP tasks confirm our approach is robust, scalable, and easily integrated with state-of-the-art deep neural networks.

## Acknowledgements

This research supported in part by the HPI Research Center in Machine Learning and Data Science at UC Irvine, as well as the UC Irvine Initiative in AI, Law, and Society.

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
