# Supplement: Scalable and Stable Surrogates for Flexible Classifiers with Fairness Constraints

**Harry Bendekgey**[*]
hbendekg@uci.edu

**Erik B. Sudderth**[*]
sudderth@uci.edu

## A   Experimental Protocols

For each experiment, we begin by observing the latent fairness discrepancy in an unconstrained model, and use that to assign group labels $a$ and $b$. In our results, we obscure these assignments and refer to the sensitive groups as $a$ and $b$ in order to reduce implicit biases associated with identity groups, as in Denton et al. [3].

### A.1   Fair Logistic Regression for Tabular Data

To preprocess each data set, we make a random $\frac{2}{3}$-$\frac{1}{3}$ train-test split and apply min-max normalization to improve regularization performance. To get an $L_2$ penalty coefficient, we perform 5-fold cross-validation on the training data and search to find a coefficient which maximizes cross-validated AUC. We searched over the set of powers of 2 from $2^{-15}$ to $2^{10}$. For Fig. 2 only, we use a penalty coefficient of $2^{-20}$ to better demonstrate the degeneracy of the surrogates.

To set the fairness penalizer $\lambda$, we searched in the range $[0, 1]$ to find a $\lambda^*$ for which $\Delta$ was slightly less than 0 on the training data set. Note that this $\lambda^*$ varies for each relaxation and fairness criterion, even on the same data set. We then ran experiments on a dense uniform grid of $\lambda$ from 0 to $\lambda^*$.

All relaxations are optimized via our Lagrangian framework. All code was implemented using PyTorch, and optimized using L-BFGS. $\lambda = 0$ models were initialized at the all-0 parameter vector, and each subsequent model was initialized starting from the solution to the previous $\lambda$ value.

We set the initial learning rate 0.1, which was chosen to achieve quick convergence on the unconstrained model. Training was terminated when every component of $\theta$ changed by $< 10^{-8}$ in a single iteration, which took less than 1 minute for every $\lambda$ on both data sets. Our code is publicly available online.[2]

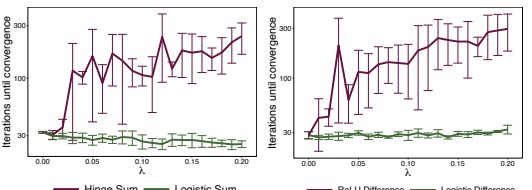

**Logistic vs Hinge Convergence.** Both Lohaus et al. [6] and Wu et al. [8] use the hinge function as their surrogates. The hinge and logistic functions have the same asymptotic behavior, but in Fig. 1 we show that Lagrangian optimization of the logistic function is quicker and more consistent due to its smoothness. We compare our logistic upper bound formulation to a hinge upper bound $g(r) = \max(0, 1 + r)$, and our logistic difference formulation to the rectified linear relaxation $g(r) = \max(0, r)$ of Zafar et al. [9]. We compare over 5 random train-test splits, with each constructed as described above. We found the optimal $L_2$ penalty coefficient independently for each split.

Figure 1: Comparison of convergence rates for logistic and piecewise linear relaxations on tabular data sets. On the left, both use upper bound regularizer $R_g^+$ to achieve demographic parity on the Adult data set. On the right, the difference framework is used to achieve equality of opportunity on COMPAS. Error bars show the standard deviation of number of iterations until convergence (changes in all parameters drop below $10^{-10}$), across 5 independent train-test splits, for the L-BFGS quasi-Newton method. For both relaxations, standard gradient descent is orders of magnitude slower.

---

[*]Department of Computer Science, University of California Irvine School of Information and Computer Science, Irvine, CA, USA

[2]https://github.com/hbendekgey/FairSurrogates

35th Conference on Neural Information Processing Systems (NeurIPS 2021).

**Adult.** We use the preprocessed data compiled by [5], which has 30,162 total data points and 100 attributes. After removing the target attribute and sensitive attributes *race*, *sex*, and *race-sex*, we further remove *capital-gain* and *capital-loss* as was done in [6]. This leaves us with $p = 94$ predictor attributes. Here we define equality of opportunity on false negative rates, i.e. predicting that someone earns under $50,000 per year when actually they earn more.

**COMPAS.** As with the Adult data set, we use the preprocessed data of [5]. We drop the target and sensitive attributes to form a data set with $p = 399$ predictive attributes and 5,273 data points. In this case, equality of opportunity is concerned with false positive rates, where a defendant who will not recidivate is incorrectly labeled high-risk.

**Toy Model.** We use binary search [6] to find a model with $\Delta = 0.03$ for each relaxation. Adding an outlier with $y = 1, s = a$ does not change the decision boundaries for any relaxation. Setting $s = b$, however, causes the linear relaxation to degenerate.

## A.2 Fair Deep Learning

For our deep learning experiments, we used the approach of Sec. A.1 to construct a range of $\lambda$ values. All results are reported on test data. All computation was done on NVIDIA GeForce RTX 2080Ti 11GB GPUs, and GPU times are reported with respect to that hardware.

**CelebA.** We used the pre-split data provided in torchvision which has 162,770 training images, 19,867 validation images and 19,962 testing images. For our architecture, we used a wide residual network (WRN-50-2) [10] initialized with random low-noise parameter values. The last layer is a soft-max layer, which is mathematically equivalent to logistic regression performed on the attributes learned from the previous layers. Thus in order to enforce fairness, the only change we make is to add the scaled fairness surrogate to the loss function. For this data set, we defined equality of opportunity on false negative rates.

The network was trained on training mini-batches of size 32 for 3 epochs. We used Adam to perform stochastic optimization with an initial learning rate of 0.01, and a scheduler which reduced the learning rate by a factor of 10 when validation loss plateaus for 2000 batches. Each epoch took 25 minutes of GPU time. With 8 relaxations and 21 $\lambda$-values per relaxation, the total GPU time was 240 hours.

**Faces of the World.** The Looking at People CVPR Challenge Track 2 [4] required participants to, given an image, return the bounding box around the face, the subject's gender, and whether or not the subject is smiling. The data set has 6,171 training images, 3,087 validation images and 8,506 test images. The participants were allowed to train on any additional data. The Faces of the World data set shows people from more varied angles than CelebA and is not limited to celebrities, a group that is not representative of the broader population in many physical or sensitive attributes.

We crop the images according to the bounding boxes provided, and resize to 224 by 224 pixels as expected by the WRN. Because the data set is small, we first trained a WRN-50-2 on CelebA using the scheme described above for our CelebA results, and then froze the first two layers of the network to prevent overfitting on Faces of the World. All experiments were initialized at this same baseline. The network was trained on mini-batches of size 32 for 30 epochs using Adam. The initial learning rate was set to 0.01 and a scheduler reduced the learning rate by a factor of 10 when validation loss plateaued for 2000 batches. Each epoch took 1 minute of GPU time. With 4 relaxations and 21 $\lambda$ values per relaxation, the total GPU time was 42 hours.

On top of having performance costs, kernel-based methods have quadratic memory requirements in the number of data points. In order to get around this, Lohaus et al. [6] construct a small set of "reasonable points" and perform learning on those. We use their publicly available code[3], converting the input images to 150,528-dimensional input vectors (224 by 224 by 3 channels) and run with as many reasonable points as we can hold in 125 gigabytes of RAM.

**Yelp.** We took the subset of reviews from the 5,000 most prolific reviewers, totaling 337,723 reviews. To estimate those reviewers' genders, we use Gender API [1] as in [7]. When no gender can be confidently inferred, we set the sensitive attribute to *unknown*.

The reweighting baseline of Calders & Verwer [2] does not propose a way weighting data with no labelled sensitive attribute. For this baseline, we simply set those points to have weight 1.

---

[3]`https://github.com/mlohaus/SearchFair`, GNU General Public License v3.0

We modified publicly-available code from Onepoint Consulting[4]. Our model was initialized to the pretrained `BertForSequenceClassification` from the Pytorch Transformers library. No parameters were frozen during our training.

The learning rate was initialized to $2 \cdot 10^{-5}$ and decreased linearly to 0 over the 3 training epochs. Weight decay with parameter $10^{-3}$ was used to improve regularization. Each epoch took 200 minutes of GPU time. To save computation time we use the same $\lambda = 0$ training session for every relaxation. With 4 relaxations and 7 $\lambda$ values per relaxation, as well as the unconstrained training session, the total GPU time was 290 hours.

## B  Conditions for Surrogate Degeneracy

For the more general case, we must consider averages of $\Phi(\mathbf{x})_j$ where we zero out negative values, as well as averages where we zero out the positive values. Define:

$$\gamma_{jsy} = \frac{1}{N_{sy}} \sum_{\substack{(\mathbf{x},s',y') \in \mathcal{D} \\ s'=s, y'=y}} \max(0, \Phi(\mathbf{x})_j), \tag{1}$$

$$\eta_{jsy} = \frac{1}{N_{sy}} \sum_{\substack{(\mathbf{x},s',y') \in \mathcal{D} \\ s'=s, y'=y}} \min(0, \Phi(\mathbf{x})_j). \tag{2}$$

Note that the normalizing $N_{sy}$ include *all* data with sensitive label $s$ and target attribute $y$, even those which the thresholding functions replace with 0. Further let $\gamma_{j.y}$ be the mean of $\Phi(\mathbf{x})_j$ (with negative values zeroed out) for data with $y' = y$ but any group, and $\gamma_{js.}$ be the mean of $\Phi(\mathbf{x})_j$ (with negative values zeroed out) for data with $s' = s$ but any target value. Define $\eta_{js.}, \eta_{j.y}$ similarly. Note that when attribute $\Phi(\mathbf{x})_j$ is non-negative, all $\eta_{jsy}$ are 0 and $\gamma_{jsy} = \mu_{jsy}$. We again assume that $g(r)$ is continuous and monotonically increasing, and that $\delta_g^+ = \lim_{r \to \infty} g'(r)$ and $\delta_g^- = \lim_{r \to -\infty} g'(r)$ exist.

**Theorem 4** (General case for degeneracy in demographic parity). *Consider a feature $j$.*
*(1) If $\delta_g^-(\gamma_{ja.} - \gamma_{jb.}) + \delta_g^+(\eta_{ja.} - \eta_{jb.}) > 0$ then for $\lambda > \lambda_j^* = \frac{\gamma_{j.1}p_{.1} - \eta_{j.0}p_{.0}}{\delta_g^-(\gamma_{ja.} - \gamma_{jb.}) + \delta_g^+(\eta_{ja.} - \eta_{jb.})}$,*
$\lim_{\mathbf{w}_j \to -\infty} \mathcal{L}_{g,\lambda}(\theta) = -\infty$.
*(2) If $\delta_g^+(\gamma_{jb.} - \gamma_{ja.}) + \delta_g^-(\eta_{jb.} - \eta_{ja.}) > 0$ then for $\lambda > \lambda_j^* = \frac{\gamma_{j.0}p_{.0} - \eta_{j.1}p_{.1}}{\delta_g^+(\gamma_{jb.} - \gamma_{ja.}) + \delta_g^-(\eta_{jb.} - \eta_{ja.})}$,*
$\lim_{\mathbf{w}_j \to +\infty} \mathcal{L}_{g,\lambda}(\theta) = -\infty$.

**Proof of Theorems 1 and 4**. We begin by proving Theorem 4. Consider the two cases separately.
(1) It is sufficient to show that as $\mathbf{w}_j \to -\infty$, $\frac{\partial}{\partial \mathbf{w}_j} \mathcal{L}_{g,\lambda}(\theta) \to \kappa > 0$.

$$\begin{aligned}
&\frac{\partial}{\partial \mathbf{w}_j} \mathcal{L}_{g,\lambda}(\theta) \\
&= \frac{\partial}{\partial \mathbf{w}_j} \left( \frac{1}{N} \sum_{(\mathbf{x},s,y) \in \mathcal{D}} \left( \mathcal{L}(\mathbf{w}^T \Phi(\mathbf{x}), y) + \lambda R_g(\mathbf{x}, y, s) \right) \right) \\
&= \frac{\partial}{\partial \mathbf{w}_j} \left( \frac{1}{N} \sum_{\substack{(\mathbf{x},s,y) \in \mathcal{D} \\ y=1}} -\log \sigma(\mathbf{w}^T \Phi(\mathbf{x})) + \frac{1}{N} \sum_{\substack{(\mathbf{x},s,y) \in \mathcal{D} \\ y=0}} -\log \sigma(-\mathbf{w}^T \Phi(\mathbf{x})) \right. \\
&\qquad \left. + \frac{\lambda}{N_{a.}} \sum_{\substack{(\mathbf{x},s,y) \in \mathcal{D} \\ s=a}} g(\mathbf{w}^T \Phi(\mathbf{x})) - \frac{\lambda}{N_{b.}} \sum_{\substack{(\mathbf{x},s,y) \in \mathcal{D} \\ s=b}} g(\mathbf{w}^T \Phi(\mathbf{x})) \right) \tag{3} \\
&= \frac{1}{N} \sum_{\substack{(\mathbf{x},s,y) \in \mathcal{D} \\ y=1}} -\sigma(-\mathbf{w}^T \Phi(\mathbf{x})) \Phi(\mathbf{x})_j + \frac{1}{N} \sum_{\substack{(\mathbf{x},s,y) \in \mathcal{D} \\ y=0}} \sigma(\mathbf{w}^T \Phi(\mathbf{x})) \Phi(\mathbf{x})_j \\
&\quad + \frac{\lambda}{N_{a.}} \sum_{\substack{(\mathbf{x},s,y) \in \mathcal{D} \\ s=a}} g'(\mathbf{w}^T \Phi(\mathbf{x})) \Phi(\mathbf{x})_j - \frac{\lambda}{N_{b.}} \sum_{\substack{(\mathbf{x},s,y) \in \mathcal{D} \\ s=b}} g'(\mathbf{w}^T \Phi(\mathbf{x})) \Phi(\mathbf{x})_j
\end{aligned}$$

---

[4] `https://github.com/onepointconsulting/yelp_bert/blob/master/bert_training.ipynb`

Note that as $\mathbf{w}_j \to -\infty$, $\sigma(-\mathbf{w}^T\Phi(\mathbf{x})) \to 0$ for data with $\Phi(\mathbf{x})_j < 0$ and $\sigma(-\mathbf{w}^T\Phi(\mathbf{x})) \to 1$ for data with $\Phi(\mathbf{x})_j > 0$. Similarly, $\sigma(\mathbf{w}^T\Phi(\mathbf{x})) \to 0$ for data with $\Phi(x)_j > 0$ and $\sigma(\mathbf{w}^T\Phi(\mathbf{x})) \to 1$ for data with $\Phi(\mathbf{x})_j < 0$. Finally, $g'(\mathbf{w}^T\Phi(\mathbf{x})) \to \delta_g^+$ for data with $\Phi(\mathbf{x})_j < 0$ and $g'(\mathbf{w}^T\Phi(\mathbf{x})) \to \delta_g^-$ for data with $\Phi(\mathbf{x})_j > 0$. Thus

$$
\begin{aligned}
\lim_{\mathbf{w}_j \to -\infty} \frac{\partial}{\partial \mathbf{w}_j} \mathcal{L}_{g,\lambda}(\theta) &= \frac{1}{N} \sum_{\substack{(\mathbf{x},s,y)\in\mathcal{D} \\ y=1}} -\Phi(\mathbf{x})_j \mathbb{1}_{\{\Phi(\mathbf{x})_j>0\}} + \frac{1}{N} \sum_{\substack{(\mathbf{x},s,y)\in\mathcal{D} \\ y=0}} \Phi(\mathbf{x})_j \mathbb{1}_{\{\Phi(\mathbf{x})_j<0\}} \\
&+ \frac{\lambda}{N_{a.}} \sum_{\substack{(\mathbf{x},s,y)\in\mathcal{D} \\ s=a}} \left( \delta_g^+ \Phi(\mathbf{x})_j \mathbb{1}_{\{\Phi(\mathbf{x})_j<0\}} + \delta_g^- \Phi(\mathbf{x})_j \mathbb{1}_{\{\Phi(\mathbf{x})_j>0\}} \right) \\
&- \frac{\lambda}{N_{b.}} \sum_{\substack{(\mathbf{x},s,y)\in\mathcal{D} \\ s=b}} \left( \delta_g^+ \Phi(\mathbf{x})_j \mathbb{1}_{\{\Phi(\mathbf{x})_j<0\}} + \delta_g^- \Phi(\mathbf{x})_j \mathbb{1}_{\{\Phi(\mathbf{x})_j>0\}} \right) \\
&= -\gamma_{j.1} p_{.1} + \eta_{j.0} p_{.0} + \lambda \left( \delta_g^+ (\eta_{ja.} - \eta_{jb.}) + \delta_g^- (\gamma_{ja.} - \gamma_{jb.}) \right) \\
&> -\gamma_{j.1} p_{.1} + \eta_{j.0} p_{.0} + \lambda_j^* \left( \delta_g^+ (\eta_{ja.} - \eta_{jb.}) + \delta_g^- (\gamma_{ja.} - \gamma_{jb.}) \right) = 0
\end{aligned}
\tag{4}
$$

Where $\lambda_j^* = \frac{\gamma_{j.1} p_{.1} - \eta_{j.0} p_{.0}}{\delta_g^- (\gamma_{ja.} - \gamma_{jb.}) + \delta_g^+ (\eta_{ja.} - \eta_{jb.})}$ is the lower bound on $\lambda$. The derivative converges to $\kappa = -\gamma_{j.1} p_{.1} + \eta_{j.0} p_{.0} + \lambda \left( \delta_g^+ (\eta_{ja.} - \eta_{jb.}) + \delta_g^- (\gamma_{ja.} - \gamma_{jb.}) \right) > 0$.

For condition (2) it is sufficient to show that as $\mathbf{w}_j \to \infty$, $\frac{\partial}{\partial \mathbf{w}_j} \mathcal{L}_{g,\lambda}(\theta) \to \kappa < 0$. Equation (3) still holds, but we instead note that as $\mathbf{w}_j \to \infty$, $\sigma(-\mathbf{w}^T\Phi(\mathbf{x})) \to 0$ for data with $\Phi(\mathbf{x})_j > 0$ and $\sigma(-\mathbf{w}^T\Phi(\mathbf{x})) \to 1$ for data with $\Phi(\mathbf{x})_j < 0$. Similarly, $\sigma(\mathbf{w}^T\Phi(\mathbf{x})) \to 0$ for data with $\Phi(\mathbf{x})_j < 0$ and $\sigma(\mathbf{w}^T\Phi(\mathbf{x})) \to 1$ for data with $\Phi(\mathbf{x})_j > 0$. Finally, $g'(\mathbf{w}^T\Phi(\mathbf{x})) \to \delta_g^+$ for data with $\Phi(\mathbf{x})_j > 0$ and $g'(\mathbf{w}^T\Phi(\mathbf{x})) \to \delta_g^-$ for data with $\Phi(\mathbf{x})_j < 0$. Thus

$$
\begin{aligned}
\lim_{\mathbf{w}_j \to \infty} \frac{\partial}{\partial \mathbf{w}_j} \mathcal{L}_{g,\lambda}(\theta) &= \frac{1}{N} \sum_{\substack{(\mathbf{x},s,y)\in\mathcal{D} \\ y=1}} -\Phi(\mathbf{x})_j \mathbb{1}_{\{\Phi(\mathbf{x})_j<0\}} + \frac{1}{N} \sum_{\substack{(\mathbf{x},s,y)\in\mathcal{D} \\ y=0}} \Phi(\mathbf{x})_j \mathbb{1}_{\{\Phi(\mathbf{x})_j>0\}} \\
&+ \frac{\lambda}{N_{a.}} \sum_{\substack{(\mathbf{x},s,y)\in\mathcal{D} \\ s=a}} \left( \delta_g^+ \Phi(\mathbf{x})_j \mathbb{1}_{\{\Phi(\mathbf{x})_j>0\}} + \delta_g^- \Phi(\mathbf{x})_j \mathbb{1}_{\{\Phi(\mathbf{x})_j<0\}} \right) \\
&- \frac{\lambda}{N_{b.}} \sum_{\substack{(\mathbf{x},s,y)\in\mathcal{D} \\ s=b}} \left( \delta_g^+ \Phi(\mathbf{x})_j \mathbb{1}_{\{\Phi(\mathbf{x})_j>0\}} + \delta_g^- \Phi(\mathbf{x})_j \mathbb{1}_{\{\Phi(\mathbf{x})_j<0\}} \right) \\
&= -\eta_{j.1} p_{.1} + \gamma_{j.0} p_{.0} + \lambda \left( \delta_g^+ (\gamma_{ja.} - \gamma_{jb.}) + \delta_g^- (\eta_{ja.} - \eta_{jb.}) \right) \\
&< -\eta_{j.1} p_{.1} + \gamma_{j.0} p_{.0} + \lambda_j^* \left( \delta_g^+ (\gamma_{ja.} - \gamma_{jb.}) + \delta_g^- (\eta_{ja.} - \eta_{jb.}) \right) = 0
\end{aligned}
\tag{5}
$$

Where $\lambda_j^* = \frac{\gamma_{j.0} p_{.0} - \eta_{j.1} p_{.1}}{\delta_g^+ (\gamma_{jb.} - \gamma_{ja.}) + \delta_g^- (\eta_{jb.} - \eta_{ja.})}$ is the lower bound on $\lambda$. The derivative converges to $\kappa = -\eta_{j.1} p_{.1} + \gamma_{j.0} p_{.0} + \lambda \left( \delta_g^+ (\gamma_{ja.} - \gamma_{jb.}) + \delta_g^- (\eta_{ja.} - \eta_{jb.}) \right) < 0$, and thus $\mathcal{L}_{g,\lambda}(\theta) \to -\infty$.

Theorem 1 is a special case of Theorem 4 where all $\eta_{jsy} = 0$, $\gamma_{jsy} = \mu_{jsy}$. The conditions in Theorem 1 make use of the fact that $\delta_g^+, \delta_g^- > 0$ from the monotonicity assumption. $\square$

**Theorem 5** (General case for degeneracy in equality of opportunity). *Consider a feature $j$.*
*(1) If $\delta_g^- (\gamma_{ja0} - \gamma_{jb0}) + \delta_g^+ (\eta_{ja0} - \eta_{jb0}) > 0$ and $\lambda > \lambda_j^* = \frac{\gamma_{j.1} p_{.1} - \eta_{j.0} p_{.0}}{\delta_g^- (\gamma_{ja0} - \gamma_{jb0}) + \delta_g^+ (\eta_{ja0} - \eta_{jb0})}$, then* $\lim_{\mathbf{w}_j \to -\infty} \mathcal{L}_{g,\lambda}(\theta) = -\infty$.
*(2) If $\delta_g^+ (\gamma_{jb0} - \gamma_{ja0}) + \delta_g^- (\eta_{jb0} - \eta_{ja0}) > 0$ and $\lambda > \lambda_j^* = \frac{\gamma_{j.0} p_{.0} - \eta_{j.1} p_{.1}}{\delta_g^+ (\gamma_{jb0} - \gamma_{ja0}) + \delta_g^- (\eta_{jb0} - \eta_{ja0})}$, then* $\lim_{\mathbf{w}_j \to +\infty} \mathcal{L}_{g,\lambda}(\theta) = -\infty$.

**Proof of Theorems 2 and 5**. The proof is identical to the proof for Theorems 1 and 4, except with the fairness metrics defined only on the negative instances. □

Finally, we present a generalization of Theorem 3:

**Theorem 6.** *For any surrogate $g(r)$ such that $\delta_g^- = 0$, both thresholds in Theorem 5 are bounded below by $p_{b0}/\delta_g^+$:*
*(1) If $\delta_g^-(\gamma_{ja0} - \gamma_{jb0}) + \delta_g^+(\eta_{ja0} - \eta_{jb0}) > 0$, then $\lambda_j^* = \frac{\gamma_{j.1}p_{.1} - \eta_{j.0}p_{.0}}{\delta_g^-(\gamma_{ja0} - \gamma_{jb0}) + \delta_g^+(\eta_{ja0} - \eta_{jb0})} \geq p_{b0}/\delta_g^+$.*
*(2) If $\delta_g^+(\gamma_{jb0} - \gamma_{ja0}) + \delta_g^-(\eta_{jb0} - \eta_{ja0}) > 0$ then $\lambda_j^* = \frac{\gamma_{j.0}p_{.0} - \eta_{j.1}p_{.1}}{\delta_g^+(\gamma_{jb0} - \gamma_{ja0}) + \delta_g^-(\eta_{jb0} - \eta_{ja0})} \geq p_{b0}/\delta_g^+$.*

**Proof of Theorems 3 and 6.** We start by proving condition (1). Applying the assumption that $\delta_g^- = 0$, we see that $\delta_g^+(\eta_{ja0} - \eta_{jb0}) > 0$. Because all $\eta_{jsy} \leq 0$ and $\delta_g^+ \geq 0$, we know $-\eta_{jb0}\delta_g^+ \geq \delta_g^+(\eta_{ja0} - \eta_{jb0}) > 0$. Thus

$$\lambda_j^* = \frac{\gamma_{j.1}p_{.1} - \eta_{j.0}p_{.0}}{\delta_g^-(\gamma_{ja0} - \gamma_{jb0}) + \delta_g^+(\eta_{ja0} - \eta_{jb0})} = \frac{\gamma_{j.1}p_{.1} - \eta_{j.0}p_{.0}}{\delta_g^+(\eta_{ja0} - \eta_{jb0})} \geq \frac{\gamma_{j.1}p_{.1} - \eta_{j.0}p_{.0}}{-\eta_{jb0}\delta_g^+} \geq \frac{-\eta_{j.0}p_{.0}}{-\eta_{jb0}\delta_g^+}$$

The last inequality uses $\gamma_{j.1}p_{.1} \geq 0$. Further note that,

$$-\eta_{j.0} = \frac{1}{N_{.0}} \sum_{\substack{(\mathbf{x},s,y)\in\mathcal{D} \\ y=0}} -\min(0, \Phi(\mathbf{x})_j) \geq \frac{1}{N_{.0}} \sum_{\substack{(\mathbf{x},s,y)\in\mathcal{D} \\ s=b,y=0}} -\min(0, \Phi(\mathbf{x})_j) = -\frac{N_{b0}}{N_{.0}}\eta_{jb0}$$

This follow from the fact that every term in these sums is non-negative. Thus

$$\lambda_j^* \geq \frac{-\eta_{j.0}p_{.0}}{-\eta_{jb0}\delta_g^+} \geq \frac{-\eta_{jb0}\frac{N_{b0}}{N_{.0}}\frac{N_{.0}}{N}}{-\eta_{jb0}\delta_g^+} = \frac{p_{b0}}{\delta_g^+}$$

Next we prove condition (2). Applying the assumption that $\delta_g^- = 0$, we see that $\delta_g^+(\gamma_{jb0} - \gamma_{ja0}) > 0$. Because all $\gamma_{jsy} \geq 0$ and $\delta_g^+ \geq 0$, we know $\gamma_{jb0}\delta_g^+ \geq \delta_g^+(\gamma_{jb0} - \gamma_{ja0}) > 0$. Thus

$$\lambda_j^* = \frac{\gamma_{j.0}p_{.0} - \eta_{j.1}p_{.1}}{\delta_g^+(\gamma_{jb0} - \gamma_{ja0}) + \delta_g^-(\eta_{jb0} - \eta_{ja0})} = \frac{\gamma_{j.0}p_{.0} - \eta_{j.1}p_{.1}}{\delta_g^+(\gamma_{jb0} - \gamma_{ja0})} \geq \frac{\gamma_{j.0}p_{.0} - \eta_{j.1}p_{.1}}{\delta_g^+\gamma_{jb0}} \geq \frac{\gamma_{j.0}p_{.0}}{\delta_g^+\gamma_{jb0}}$$

The last inequality uses $-\eta_{j.1}p_{.1} \geq 0$. Further note that

$$\gamma_{j.0} = \frac{1}{N_{.0}} \sum_{\substack{(\mathbf{x},s,y)\in\mathcal{D} \\ y=0}} \max(0, \Phi(\mathbf{x})_j) \geq \frac{1}{N_{.0}} \sum_{\substack{(\mathbf{x},s,y)\in\mathcal{D} \\ s=b,y=0}} \max(0, \Phi(\mathbf{x})_j) = \frac{N_{b0}}{N_{.0}}\gamma_{jb0}$$

This follows from the fact that every term in these sums is non-negative. Thus

$$\lambda_j^* \geq \frac{\gamma_{j.0}p_{.0}}{\delta_g^+\gamma_{jb0}} \geq \frac{\gamma_{jb0}\frac{N_{b0}}{N_{.0}}\frac{N_{.0}}{N}}{\delta_g^+\gamma_{jb0}} = \frac{p_{b0}}{\delta_g^+}$$

Theorem 3 is a special case of Theorem 6 where $\Phi(\mathbf{x})_j \geq 0$ so the thresholds in Theorem 5 are given by the simpler equations in Theorem 2. As we have shown that the thresholds in Theorem 5 are bounded, the equivalent but simplified thresholds in Theorem 2 must be as well. □

Finally, we generalize Thm. 4 to allow for the use of two independent surrogates: $g_0$ applied to data with target label $y = 0$, and $g_1$ applied to data with target label $y = 1$. This applies to our use of the log-sigmoid difference surrogate on the demographic parity criterion. The regularizer is given by

$$R_{g_0,g_1}(\mathbf{x}, y, s) = \begin{cases} \frac{g_0(f_\theta(\mathbf{x}))}{p_{a.}} & \text{if } s = a, y = 0 \\ \frac{g_1(f_\theta(\mathbf{x}))}{p_{a.}} & \text{if } s = a, y = 1 \\ -\frac{g_0(f_\theta(\mathbf{x}))}{p_{b.}} & \text{if } s = b, y = 0 \\ -\frac{g_1(f_\theta(\mathbf{x}))}{p_{b.}} & \text{if } s = b, y = 1. \end{cases} \tag{6}$$

With overall loss function

$$\mathcal{L}_{g_0,g_1,\lambda}(\theta) = \frac{1}{N} \sum_{(\mathbf{x},s,y)\in\mathcal{D}} \left( \mathcal{L}(\mathbf{w}^T\Phi(\mathbf{x}), y) + \lambda R_{g_0,g_1,\lambda}(\mathbf{x}, y, s) \right) \tag{7}$$

For this proof, we must further define $p_{y|s} = \frac{N_{sy}}{N_{s.}}$. We assume that both $g_0(r), g_1(r)$ are continuous and monotonically increasing, and that $\delta_{g_0}^+ = \lim_{r\to\infty} g_0'(r)$, $\delta_{g_1}^+ = \lim_{r\to\infty} g_1'(r)$, $\delta_{g_0}^- = \lim_{r\to-\infty} g_0'(r)$ and $\delta_{g_1}^- = \lim_{r\to-\infty} g_1'(r)$ exist.

Finally, for notational compactness, we write our thresholds' denominators:

$$\beta^- = \delta_{g_0}^+(p_{0|a}\eta_{a0} - p_{0|b}\eta_{b0}) + \delta_{g_1}^+(p_{1|a}\eta_{a1} - p_{1|b}\eta_{b1}) + \delta_{g_0}^-(p_{0|a}\gamma_{a0} - p_{0|b}\gamma_{b0}) + \delta_{g_1}^-(p_{1|a}\gamma_{a1} - p_{1|b}\gamma_{b1})$$

$$\beta^+ = \delta_{g_0}^+(p_{0|b}\gamma_{b0} - p_{0|a}\gamma_{a0}) + \delta_{g_1}^+(p_{1|b}\gamma_{b1} - p_{1|a}\gamma_{a1}) + \delta_{g_0}^-(p_{0|b}\eta_{b0} - p_{0|a}\eta_{a0}) + \delta_{g_1}^-(p_{1|b}\eta_{b1} - p_{1|a}\eta_{a1})$$

**Theorem 7** (Two-surrogate case for degeneracy in demographic parity). *Consider a feature $j$.*

*(1) If $\beta^- > 0$ then for $\lambda > \lambda_j^* = \frac{\gamma_{j.1}p_{.1} - \eta_{j.0}p_{.0}}{\beta^-}$, $\lim\limits_{\mathbf{w}_j \to -\infty} \mathcal{L}_{g,\lambda}(\theta) = -\infty$.*

*(2) If $\beta^+ > 0$ then for $\lambda > \lambda_j^* = \frac{\gamma_{j.0}p_{.0} - \eta_{j.1}p_{.1}}{\beta^+}$, $\lim\limits_{\mathbf{w}_j \to +\infty} \mathcal{L}_{g,\lambda}(\theta) = -\infty$.*

**Proof of Theorem 7.** The proof proceeds similarly as Theorem 4. First we handle (1) by proving that as $\mathbf{w}_j \to -\infty$, $\frac{\partial}{\partial \mathbf{w}_j}\mathcal{L}_{g,\lambda}(\theta) \to \kappa > 0$.

$$\frac{\partial}{\partial \mathbf{w}_j}\mathcal{L}_{g_0,g_1,\lambda}(\theta)$$

$$= \frac{\partial}{\partial \mathbf{w}_j}\left( \frac{1}{N}\sum_{\substack{(\mathbf{x},s,y)\in\mathcal{D}\\y=1}} -\log\sigma(\mathbf{w}^T\Phi(\mathbf{x})) + \frac{1}{N}\sum_{\substack{(\mathbf{x},s,y)\in\mathcal{D}\\y=0}} -\log\sigma(-\mathbf{w}^T\Phi(\mathbf{x})) \right.$$

$$+ \frac{\lambda}{N_{a.}}\sum_{\substack{(\mathbf{x},s,y)\in\mathcal{D}\\s=a,y=0}} g_0(\mathbf{w}^T\Phi(\mathbf{x})) + \frac{\lambda}{N_{a.}}\sum_{\substack{(\mathbf{x},s,y)\in\mathcal{D}\\s=a,y=1}} g_1(\mathbf{w}^T\Phi(\mathbf{x}))$$

$$\left. - \frac{\lambda}{N_{b.}}\sum_{\substack{(\mathbf{x},s,y)\in\mathcal{D}\\s=b,y=0}} g_0(\mathbf{w}^T\Phi(\mathbf{x})) - \frac{\lambda}{N_{b.}}\sum_{\substack{(\mathbf{x},s,y)\in\mathcal{D}\\s=b,y=1}} g_1(\mathbf{w}^T\Phi(\mathbf{x})) \right) \qquad (8)$$

$$= \frac{1}{N}\sum_{\substack{(\mathbf{x},s,y)\in\mathcal{D}\\y=1}} -\sigma(-\mathbf{w}^T\Phi(\mathbf{x}))\Phi(\mathbf{x})_j + \frac{1}{N}\sum_{\substack{(\mathbf{x},s,y)\in\mathcal{D}\\y=0}} \sigma(\mathbf{w}^T\Phi(\mathbf{x}))\Phi(\mathbf{x})_j$$

$$+ \frac{\lambda}{N_{a.}}\sum_{\substack{(\mathbf{x},s,y)\in\mathcal{D}\\s=a,y=0}} g_0'(\mathbf{w}^T\Phi(\mathbf{x}))\Phi(\mathbf{x})_j + \frac{\lambda}{N_{a.}}\sum_{\substack{(\mathbf{x},s,y)\in\mathcal{D}\\s=a,y=1}} g_1'(\mathbf{w}^T\Phi(\mathbf{x}))\Phi(\mathbf{x})_j$$

$$- \frac{\lambda}{N_{b.}}\sum_{\substack{(\mathbf{x},s,y)\in\mathcal{D}\\s=b,y=0}} g_0'(\mathbf{w}^T\Phi(\mathbf{x}))\Phi(\mathbf{x})_j - \frac{\lambda}{N_{b.}}\sum_{\substack{(\mathbf{x},s,y)\in\mathcal{D}\\s=b,y=1}} g_1'(\mathbf{w}^T\Phi(\mathbf{x}))\Phi(\mathbf{x})_j$$

We observe, as in the proof for Theorem 4, that $\sigma(\mathbf{w}^T\Phi(\mathbf{x}))$ becomes 0 or 1 as $\mathbf{w}_j \to \pm\infty$, depending on the sign of $\Phi(\mathbf{x})_j$

$$\lim_{\mathbf{w}_j \to -\infty} \frac{\partial}{\partial \mathbf{w}_j}\mathcal{L}_{g_0,g_1,\lambda}(\theta) = \frac{1}{N}\sum_{\substack{(\mathbf{x},s,y)\in\mathcal{D}\\y=1}} -\Phi(\mathbf{x})_j 1_{\{\Phi(\mathbf{x})_j>0\}} + \frac{1}{N}\sum_{\substack{(\mathbf{x},s,y)\in\mathcal{D}\\y=0}} \Phi(\mathbf{x})_j 1_{\{\Phi(\mathbf{x})_j<0\}}$$

$$+ \frac{\lambda}{N_{a.}}\sum_{\substack{(\mathbf{x},s,y)\in\mathcal{D}\\s=a,y=0}} \left( \delta_{g_0}^+\Phi(\mathbf{x})_j 1_{\{\Phi(\mathbf{x})_j<0\}} + \delta_{g_0}^-\Phi(\mathbf{x})_j 1_{\{\Phi(\mathbf{x})_j>0\}} \right)$$

$$+ \frac{\lambda}{N_{a.}}\sum_{\substack{(\mathbf{x},s,y)\in\mathcal{D}\\s=a,y=1}} \left( \delta_{g_1}^+\Phi(\mathbf{x})_j 1_{\{\Phi(\mathbf{x})_j<0\}} + \delta_{g_1}^-\Phi(\mathbf{x})_j 1_{\{\Phi(\mathbf{x})_j>0\}} \right)$$

$$- \frac{\lambda}{N_{b.}}\sum_{\substack{(\mathbf{x},s,y)\in\mathcal{D}\\s=b,y=0}} \left( \delta_{g_0}^+\Phi(\mathbf{x})_j 1_{\{\Phi(\mathbf{x})_j<0\}} + \delta_{g_0}^-\Phi(\mathbf{x})_j 1_{\{\Phi(\mathbf{x})_j>0\}} \right) \qquad (9)$$

$$- \frac{\lambda}{N_{b.}}\sum_{\substack{(\mathbf{x},s,y)\in\mathcal{D}\\s=b,y=1}} \left( \delta_{g_1}^+\Phi(\mathbf{x})_j 1_{\{\Phi(\mathbf{x})_j<0\}} + \delta_{g_1}^-\Phi(\mathbf{x})_j 1_{\{\Phi(\mathbf{x})_j>0\}} \right)$$

$$= -\gamma_{j.1}p_{.1} + \eta_{j.0}p_{.0} + \lambda\beta^-$$

$$> -\gamma_{j.1}p_{.1} + \eta_{j.0}p_{.0} + \lambda_j^*\beta^- = 0$$

Where $\lambda_j^* = \frac{\gamma_{j.1}p_{.1} - \eta_{j.0}p_{.0}}{\beta^-}$ is the lower bound on $\lambda$. The derivative converges to $\kappa = -\gamma_{j.1}p_{.1} + \eta_{j.0}p_{.0} + \lambda\beta^- > 0$.

For condition (2) it is sufficient to show that as $\mathbf{w}_j \to \infty$, $\frac{\partial}{\partial \mathbf{w}_j}\mathcal{L}_{g,\lambda}(\theta) \to \kappa < 0$. Equation (8) still holds, and we proceed:

$$
\begin{aligned}
\lim_{\mathbf{w}_j \to \infty} \frac{\partial}{\partial \mathbf{w}_j}\mathcal{L}_{g_0,g_1,\lambda}(\theta) = {} & \frac{1}{N}\sum_{\substack{(\mathbf{x},s,y)\in\mathcal{D}\\y=1}} -\Phi(\mathbf{x})_j 1_{\{\Phi(\mathbf{x})_j<0\}} + \frac{1}{N}\sum_{\substack{(\mathbf{x},s,y)\in\mathcal{D}\\y=0}} \Phi(\mathbf{x})_j 1_{\{\Phi(\mathbf{x})_j>0\}} \\
& + \frac{\lambda}{N_{a.}}\sum_{\substack{(\mathbf{x},s,y)\in\mathcal{D}\\s=a,y=0}} \left(\delta_{g_0}^+ \Phi(\mathbf{x})_j 1_{\{\Phi(\mathbf{x})_j>0\}} + \delta_{g_0}^- \Phi(\mathbf{x})_j 1_{\{\Phi(\mathbf{x})_j<0\}}\right) \\
& + \frac{\lambda}{N_{a.}}\sum_{\substack{(\mathbf{x},s,y)\in\mathcal{D}\\s=a,y=1}} \left(\delta_{g_1}^+ \Phi(\mathbf{x})_j 1_{\{\Phi(\mathbf{x})_j>0\}} + \delta_{g_1}^- \Phi(\mathbf{x})_j 1_{\{\Phi(\mathbf{x})_j<0\}}\right) \\
& - \frac{\lambda}{N_{b.}}\sum_{\substack{(\mathbf{x},s,y)\in\mathcal{D}\\s=b,y=0}} \left(\delta_{g_0}^+ \Phi(\mathbf{x})_j 1_{\{\Phi(\mathbf{x})_j>0\}} + \delta_{g_0}^- \Phi(\mathbf{x})_j 1_{\{\Phi(\mathbf{x})_j<0\}}\right) \\
& - \frac{\lambda}{N_{b.}}\sum_{\substack{(\mathbf{x},s,y)\in\mathcal{D}\\s=b,y=1}} \left(\delta_{g_1}^+ \Phi(\mathbf{x})_j 1_{\{\Phi(\mathbf{x})_j>0\}} + \delta_{g_1}^- \Phi(\mathbf{x})_j 1_{\{\Phi(\mathbf{x})_j<0\}}\right) \\
= {} & -\eta_{j.1}p_{.1} + \gamma_{j.0}p_{.0} + \lambda(-\beta^+) \\
< {} & -\eta_{j.1}p_{.1} + \gamma_{j.0}p_{.0} + \lambda_j^*(-\beta^+) = 0
\end{aligned}
\tag{10}
$$

Where $\lambda_j^* = \frac{\gamma_{j.0}p_{.0} - \eta_{j.1}p_{.1}}{\beta^+}$ is the lower bound on $\lambda$. The derivative converges to $\kappa = -\eta_{j.1}p_{.1} + \gamma_{j.0}p_{.0} - \lambda\beta^+ < 0$ and thus $\mathcal{L}_{g_0,g_1,\lambda}(\theta) \to -\infty$. $\qquad\square$