# OpenReview forum: "Scalable and Stable Surrogates for Flexible Classifiers with Fairness Constraints"
_NeurIPS.cc/2021/Conference — NeurIPS 2021 Poster_

### Official Review · Reviewer_T3F6 · 2021-07-11

**Rating:** 7
**Confidence:** 4

**Summary:**

The paper discusses several relaxations for the hard-indicator based fairness measures and compares their fairness-accuracy tradeoffs.

Detailed discussion on the necessary conditions for the failure case of certain relaxations (like linear or hinge) is provided. When the said conditions are satisfied, it is shown that the method converges to degenerate solutions where improving the fairness measure results in very high loss values of the prediction function.

Three surrogates are proposed, namely, the logistic sum, the logistic difference, and the sigmoid relaxation. They are shown to be more stable than the linear and hinge relaxations especially in non-convex models.

Experimental results are provided on tabular, vision and NLP datasets.

**Limitations And Societal Impact:**

Authors have discussed the limitations of their work.

**Main Review:**

## Strengths

The paper identifies that degenerate solutions are not caused not by the failure of KKT conditions but instead because of the conditions discovered in the paper (one of the important ones being the asymptotic behaviour). As demonstrated in the paper, despite \lambda being bounded, the Lagrangian could turn out to be unbounded.

The logistic sum relaxation offers a means to search for only one parameter ($\lambda$)  instead of two ($\lambda$ and $\tau$). Here, $\lambda$ is the constraint multiplier and $\tau$ appears in DDP $\le \tau$. Varying only the value of $\lambda$ for a feasible solution of the objective function ensures that the tolerance is brought under $\epsilon$.

It is a strength that the claims of the paper on the degeneracy of current fairness relaxations is supported both by empirical and theoretical evidence (Theorem 1, 2). The experiments support the applicability of the proposed relaxation even in the regime of non-convex models such as deep networks.

Section 5 offers some valuable insights into the behavior of the relaxations, such as -- despite the similar asymptotic behavior of ReLU and the proposed logistic difference, degeneracy is observed only with ReLU indicating the importance of adopting smooth relaxations.


## Weaknesses

Degeneracy is primarily caused by the fairness measure $\Delta_g$ approaching $-\infty$. This happens when the parameters of the model $w$ diverge. This is as per Theorem 1 and 2. A simpler alternative to prevent degeneracy would be to incorporate weight clipping / gradient clipping approaches. It is unclear how such regularization techniques compare against the use of the proposed relaxations.

The assumption for degeneracy $\delta_g \ne 0$ is weak as only a handful of relaxations, such as max(x, 0) or a linear function, qualify for this criterion. The use of softmax and gaussian functions, with $\delta_g = 0$, as a relaxation has been very common in the literature. Eg., eq (2) in  [1] or [2]. Very few methods in the existing literature have adopted max(x, 0) or linear relaxation, thereby reducing the relevance of the paper within the community.

To prepare the claims on degeneracy of fairness relaxations, the authors use the inequality constraint $\Delta_g \le \epsilon$ rather than $\Delta_g = 0$ adopted by several prior works (section 3.1 of the paper). It is unclear what motivates this modeling choice. Observe that adopting $\Delta_g = 0$ has an immediate benefit -- it would prevent the failure case where $\Delta_g$ is significantly negative yet the DDP measure is not satisfied.

The paper misses out on discussing generalization ability of the proposed relaxation techniques. It is well studied [3] that a differentiable surrogate to the indicator function leads to over-constrained objective, causing poor generalization performance. It is unclear if any of the proposed relaxations suffer from such an over-constraining phenomena. It is further unclear how the gap between the training and test accuracies differs across different relaxations.


[1] http://proceedings.mlr.press/v28/zemel13.pdf

[2] https://arxiv.org/pdf/1412.5244.pdf

[3] https://www.fatml.org/media/documents/training_fairness_constrained_classifiers_to_generalize.pdf



## Comments / Questions

Can the highly unstable training, seen on CelebA and FOTW datasets due to the use of linear relaxation, be reduced by lowering the learning rate of the model?

line 43 in the paper argues that the proposed surrogates are robust to outliers. However, there is no experimental or theoretical evidence provided for this in the paper. Please elaborate or edit the line accordingly.




**Time Spent Reviewing:**

8-10

---

> ### Author Response · Authors · 2021-08-10
> **Response to Reviewer 4**
>
> Thank you for your praise, thorough feedback, and questions. We appreciate your thorough summary of the strengths of our paper, and we hope to address your concerns.
>
> - The degeneracy of the linear relaxation is not caused by the optimization routine ending up in a region of parameter space with poor objective values; it is caused by it ending up in a region with ideal objective values (diverging to $-\infty$). Weight clipping, gradient clipping, or learning rate tuning would, at best, prevent the learning algorithm from optimizing its own objective, which calls into question the choice of objective function and stops us from accessing the Lagrangian theory behind this method, because we are no longer aiming for the optimum of our objective.
>
> - We disagree with the assertion that linear or $\max(0,r)$ surrogates have rarely been adopted in the literature. Recent papers on in-processing fairness-constrained learning such as [1] and [2] require surrogates but either do not discuss the choice of surrogate or claim their method works with any surrogate. Both of these papers use $g(r)=\max(0,1+r)$ in their code. Thus, we feel a thorough investigation into surrogate choice and their properties is an important addition to the scientific discourse. The two citations you provide are fair representation learning papers, which are part of an intimately-related but distinct line of work.
>
> - Different practitioners have different unfairness tolerances. Fixing $\Delta_g=0$ removes this flexibility, and furthermore lacks any guarantees for the true fairness discrepancy $\Delta$. For example, an unconstrained model on the toy data set in Fig. 1 (including the outlier) has $\Delta > 0$, but $\Delta_g < 0$. Thus bringing $\Delta_g=0$ would actually make unfairness worse. We briefly describe our choice on lines 151-153. For a thorough investigation into the problems of fixing $\Delta_g=0$, see [1].
>
> - In Fig. 3, train and test results for each relaxation are plotted next to one another. Although we do not provide a one-to-one correspondence of points on the training curve to points on the test curve, we can still see there is no dramatic difference in generalizability between the relaxations.
>
> - We will remove the line about the outlier. The only example of this robustness currently in the paper is Fig. 1.
>
>
> [1] Lohaus et al., Too relaxed to be fair (2020)
> [2] Cotter et al., Two-player games for efficient non-convex constrained optimization (2019)

---

> > ### Comment · Reviewer_T3F6 · 2021-08-30
> > **Response to the Authors**
> >
> > I thank the authors for their comments and clarifications. The point regarding the constraint on $\Delta_g$ being an inequality is convincing. For the point on generalizability of the proposed relaxation techniques, it would be helpful to provide more evidence theoretical or empirical in the updated version of the paper. Overall, the paper is thorough and insightful, therefore I recommend a rating of "7" to the Area Chair.

---

### Official Review · Reviewer_4qDK · 2021-07-15

**Rating:** 6
**Confidence:** 4

**Summary:**

The authors study fairness relaxations, especially in deep learning settings. They propose several conditions under which prior fairness surrogates may degenerate. Three new surrogates are proposed further to avoid degeneracy and the performance and stability of the new surrogates are proved by real-world experiments.

**Limitations And Societal Impact:**

As shown in the main review part, theory, experiment, and writing could be improved.

**Main Review:**

Generally, the authors propose several interesting fairness relaxations and prove the effectiveness both theoretically and empirically. However, more analysis on the theories is needed and the experiment results should be further explained. Meanwhile, the writing of the paper should be improved.

**Pros**

1. Theory part: Theorem 1 and 2 are insightful results and they show that under some conditions, ill-defined fairness relaxations will degenerate. Meanwhile, it is interesting to develop the logistic sum from the logistic difference.
2. Experiment part: Different kinds of datasets are included (both tabular and deep tasks). In addition, several meaningful conjectures are proposed, which may shed light on the optimization of fairness in deep models.

**Cons**

1. Theory part: Theorem 1 and 2 propose the critical points. However, it will be more useful if the authors could discuss the tightness of the critical point. In other words, when will the critical points be small enough so that the relaxations would totally fail and when will the critical points be large?
2. Experiment part: An important baseline [1] is missing for the DP setting. In addition, improvements over ReLU Difference for the EO setting are not significant.
3. Writing part: The writing should be improved. Some notations are used without explanation (such as $\gamma$ at Line 283) and some notations are introduced in different parts of the paper so that it is difficult to look up (such as $p_{s0}$ and $\mu_{jsy}$). I think it would be better if the authors could provide a paragraph to introduce all the notations solely. In addition, explanations of the experiments on tabular datasets are missing. This may be caused by the space limit but some texts could be moved to the appendix (such as Line 317 to 338).
4. Some small typos:
   - Line 249: "A generalization Theorem 3" --> "A generalization of Theorem 3"
   - Line 342: "to induce cause degeneracy as in Theorems 1-2" --> "to cause" or "to induce"
   - Figure 4 subcaption: "CelebA (EOO)" --> "CelebA (EO)"

**References**

[1] Zafar, Muhammad Bilal, et al. "Fairness constraints: Mechanisms for fair classification." Artificial Intelligence and Statistics. PMLR, 2017.

**Time Spent Reviewing:**

5

---

> ### Author Response · Authors · 2021-08-10
> **Response to Reviewer 3**
>
> Thank you for your thorough and constructive feedback. We have fixed the typos you identified, and improved clarity and notation as per your feedback. We hope the following clarifications address your theoretical and experimental concerns.
>
> 1. The threshold is small primarily when the denominator is large, meaning that the attribute takes on very different average values for the two sensitive groups. However, we feel that the important observations about these bounds are (1) they are small enough in practice to induce degeneracy on two of the most popular tabular fairness data sets, which we demonstrate, and (2) a neural network can learn representations so as to minimize these critical points (and thus bring the objective to $-\infty$). This is described briefly on lines 212-214. A neural network with sufficient capacity can bring $\lambda^*$ arbitrarily close to 0 for the linear relaxation (and thus diverge for any $\lambda > 0$), while Thm 3 tells us that for the ReLU difference and logistic difference formulations, $\lambda^*$ can only go as low as $p_{b0}$. Our experiments validate this theory.
>
> 2. The citation you provide is citation [49] in our paper. That work uses a linear relaxation, which we compare to in all our DP experiments. Regarding the performance of the ReLU difference surrogate, we emphasize the bottom right of Fig. 3, where it diverges on a popular tabular data set while our proposed alternatives remain stable. Discussion of this is included on lines 345-350.
>
> 3. Thank you again for the suggestions to improve clarity! We have moved around notation and removed references to $\gamma$, which was a hyper parameter in Wu et al.’s work. We would like to point out that detailed explanations of the experiments on tabular data are included in the appendix, Sec. A.1. Due to the popularity of COMPAS and Adult in the fairness literature, we felt it was more important to include descriptions of the text and image data sets in the main paper.

---

> > ### Comment · Reviewer_4qDK · 2021-08-19
> > **RE: Author Response**
> >
> > Thanks for the authors' response. The authors addressed my theoretical concern and clarified my misunderstandings in the experiments part. I will increase the rating to 6. And I want to see more improvements in writing in the final version of the paper. Some points I list as follow:
> >
> > 1. The references of baselines should be introduced more clearly in the experiments part. I missed baseline [49] when I first read the paper.
> > 2. The range of x-axis in "Adult-DEO" setting in Figure 3 should be changed so that the figures could be more clear.

---

> > > ### Author Response · Authors · 2021-08-23
> > > **Follow-Up to Reviewer 3**
> > >
> > > Thank you for your response. We will add citations to the experiments section of the paper to improve clarity.
> > > Note that in Fig. 3, we use the same horizontal axis values for DDP and DEO (chosen to accommodate the wider of the two) to highlight when some fairness criterion can be achieved with very little change in predictive performance.

---

### Official Review · Reviewer_C1h7 · 2021-07-17

**Rating:** 6
**Confidence:** 3

**Summary:**

The authors study several surrogates for fairness constraints. They show that simple surrogates such as linear can lead to degeneracy. They show that when the derivative of the surrogate does not go to zero as the prediction goes to infinite, we should deal with degeneracy. They show that such degeneracy is not only theoretical and happens on real-world datasets as well for linear surrogates.

**Limitations And Societal Impact:**

yes

**Main Review:**

The authors study several surrogates for fairness constraints. They show that simple surrogates such as linear can lead to degeneracy. They show that when the derivative of the surrogate does not go to zero as the prediction goes to infinite, we should deal with degeneracy. They show that such degeneracy is not only theoretical and happens on real-world datasets as well for linear surrogates.

The writing can be improved, I had a hard time understanding some parts of the paper! Especially some parts of the paper were really dense and it was hard to understand the main point.

People usually use the loss function as the surrogate for f especially when working with DNN. So, the proposed surrogates (sigmoid and log sigmoid) are not very novel. However, the thorough analysis of them and showing that they will not lead to degeneracy was interesting!

Minimizing the worst-case group loss (either grouping data points with label and sensitive attribute to achieve EOO or only sensitive attribute to achieve DP) is also a very common technique for fairness application (e..g, group-DRO). (note that we can always randomize the performance of the better group if equality is really the goal!). Do they also have the mentioned degeneracy problems?
I didn’t exactly follow why Agrawal et al. method may fail, I think a better description there would help.


**Time Spent Reviewing:**

3

---

> ### Author Response · Authors · 2021-08-10
> **Response to Reviewer 2**
>
> Thank you for your praise and constructive feedback. To improve the clarity of the writing, we have moved around the introduction of some notation so it can be easily referenced later in the paper. If there are specific (sub)sections that you feel were unclear in their relation to the main point of the paper, we would welcome that feedback.
>
> To our knowledge, the degeneracy problems we identify do not apply to the practice of minimizing the worst-case group loss. However, that methodology has separate weaknesses. Its ultimate goal is to achieve unfairness of 0, compared to our goal of bringing it below some strictly positive unfairness tolerance $\epsilon$ (see lines 65-69 for our discussion of the difference). Thus, it lacks the flexibility for a practitioner to tune the model based on how strict their fairness requirements are. Further, it lacks any guarantees of true fairness, unlike our upper bound surrogates. As you point out, you can randomize the classifier on one group to close that gap, but that induces disparate treatment, which we argue against on lines 90-95.

---

### Official Review · Reviewer_okm9 · 2021-07-18

**Rating:** 6
**Confidence:** 4

**Summary:**

A popular approach in fair machine learning in-processing methods is to replace the original fairness constraints, which are non-convex and non-differentiable, with surrogates that are easier to handle. This paper studies several of these relaxations. On the one hand, it shows some of their limits by proving sufficient conditions under which existing surrogates may lead to optimization problems where the Lagrangian has no stationary points. On the other hand, it proposes three relaxations that are less prone to this undesirable behavior.

**Limitations And Societal Impact:**

The limitations and potential negative societal impact of this work were correctly addressed.

**Main Review:**

Pros:
 - The paper is relatively easy to follow
 - Studying the limits of surrogates in fair machine learning is an important problem
 - The paper comes with both theoretical and empirical results

Cons:
 - Some comparison to existing work is missing
 - It remains unclear how fair models can be efficiently found using the proposed relaxations
 - The theoretical results are limited to very specific problems

Detailed comments:
1. Surrogate constraints are an appealing way to enforce fairness in machine learning as they are relatively easy to implement and the resulting constrained optimization problems may be solved efficiently. Unfortunately, these surrogates are often not guaranteed to lead to fair solutions. This paper proposes to study a particular case of failure where the lagrangian of the problem may become arbitrarily small, potentially leading to very sub-optimal solutions. Thus, proposing relaxations that can avoid this pitfall is important.

2. A comparison with the work of Cotter et al., Two-player games for efficient non-convex constrained optimization (2019) is missing. It studies the more general problem of non-convex constrained optimization and proposes an approach based on surrogates in a lagrangian formulation that is very reminiscing of the problem considered here. This approach should, at the very least, be considered as a baseline in the experiments

3. While it is interesting to exhibit sufficient conditions under which surrogates may lead to degeneracies (Theorem 1 and 2), it remains unclear whether the relaxations will behave well when the conditions are not met. Furthermore, the result seems to be limited to linear models (since the representation is fixed) with a particular family of loss functions. This greatly reduces the interest of the results.

4. Three different surrogates are proposed with the common idea that the indicator function should be relaxed. While the relaxations themselves are not new, for example it has long been known that the sigmoid may be used to approximate the indicator function, the paper gives interesting insights on their particular interest for fairness and gives interesting connections to the existing literature.

5. From the paper, it seems that a critical point to achieve fairness using the proposed relaxations is to correctly choose the Lagrangians parameters \lambda. Unfortunately, the paper does not provide any algorithm to find the optimal parameters in an efficient way (a grid-search is used in the experiments). This might not be trivial in practice and thus slightly reduces the overall interest of the paper.

Summary of the review:

On the one hand, this paper studies the important problem of surrogates in fair machine learning showing the limits of existing formulations and proposing alternative relaxations. On the other hand, the applicability of the theoretical results remains quite limited. Furthermore, a comparison with the work of Cotter et al., that seems quite close in spirit, is missing. Overall, I think that this paper presents valuable ideas but is slightly below the acceptance threshold.

**Time Spent Reviewing:**

6

---

> ### Author Response · Authors · 2021-08-10
> **Response to Reviewer 1**
>
> Thank you for your praise and for your constructive feedback. We hope the following clarifications address your concerns.
>
> 2 and 5. Thank you for the citation to Cotter et al.’s work. We have added a discussion of their work to Section 3.3. However, we believe that their approach does not make sense as a baseline in our experiments, because their work is complementary to ours. Their paper provides an alternative to our grid search for tuning the fairness hyperparameter lambda, but they require the use of a surrogate. They do not identify or discuss their choice, and their code uses $g(r) = \max(0,1+r)$, one of the surrogates that exhibit degeneracy in Fig. 2. In contrast, our work focuses on comparing surrogates and their properties, an investigation which we feel is missing from the scientific discourse. We further note that the degeneracy and instability of some relaxations are most clearly illustrated by our evaluation on a dense grid of $\lambda$ values. For a practitioner who is concerned with computational speed, insights from our paper can be combined with existing dual ascent or Lagrangian algorithms such as Cotter et al.’s.
>
> 3. Although Theorems 1-2 are formulated with respect to a fixed representation $\Phi(x)$, they extend easily to neural network training where $\lambda$ is fixed but $\Phi(x)$ is learned. This is briefly described on lines 212-214. If any learnable representation induces a critical $\lambda^* < \lambda$, then the subset of parameter space which defines that representation has objective values unbounded below and thus the whole objective is unbounded below. We see this experimentally: even for small values of $\lambda$ the linear relaxation finds a representation where the objective is unbounded below by minimizing $\lambda^*$ (see Fig. 4).
>
> Regarding Thms 1-2 being sufficient and not necessary conditions, we would like to emphasize Fig. 2. For the two cases of degeneracy observed on tabular data, the theoretical results from Thms 1-2 predict with high accuracy the point at which the surrogate diverges without change to the true unfairness. Further, for the deep learning models (Fig. 4) our sigmoid and logistic sum surrogates cannot degenerate according to Thms 1-2, and Thm 3 tells us the difference surrogates can only degenerate for $\lambda > p_{b0}$, unlike the linear which can degenerate for any $\lambda > 0$. Our observations match this theory: the linear surrogate is unstable for any $\lambda > 0$, while the other relaxations are not. Although we do not prove there are no other conditions for degeneracy, our wide range of experiments provide evidence that Thms 1-2 are noteworthy.

---

> > ### Comment · Reviewer_okm9 · 2021-08-23
> > **Post-rebuttal comments**
> >
> > Thank you for this rebuttal. It addressed most of my comments and only a few concerns remain.
> >
> > Comment 1 and 4: Did not need to be addressed.
> >
> > Comment 2 and 5: The rebuttal mostly addressed these two comments. Nevertheless, I remain convinced that adding this method as a baseline would make the experiments more convincing.
> >
> > Comment 3: Thanks for pointing out the remark Line 212. This addressed the second part of my comment. Regarding the first part of my comment, the theory does not guarantee that a surrogate will not degenerate when the conditions are not met. Hence, my concern remains, even if the empirical evidence tends to show that this is not likely to happen.
> >
> > Since most of my concerns were addressed and the paper has valuable contributions, I increased my score to 6.

---

### Author Response · Authors · 2021-08-10
**General Response to Reviewers**

Thank you to all reviewers for your thoughtful and thorough feedback. We have made changes to the paper to improve clarity and notation as requested. We believe some of the conceptual concerns stem from misunderstandings of our results, and have focused our individual responses on clarifying the scope of the theorems and their connections to related work.

---

### Decision · Program_Chairs · 2021-09-27

**Decision:**

Accept (Poster)

**Comment:**

The paper studies the effect of different fairness relaxations in the training of "fair" classifiers. The reviewers point out some limitations of the work, but think that the paper has ideas that could be interesting to the NeurIPS community, and therefore recommend acceptance. The authors are strongly encouraged to incorporate the feedback provide, and in particular, work on improving the writing as suggested by the reviewers.

**Additional comments from AC:**

To Reviewer okm9's comment on comparison to Cotter et al., my understanding of their work is that they propose a Lagrangian-based solver, in which the classifier is optimized using surrogate relaxations, whereas the Lagrange multipliers or costs $\lambda$ are updated using the *unrelaxed fairness constraint*. I think this is reminiscent of how you perform grid search, although when there are multiple constraints (e.g. over large number of protected groups), a simple grid search may not be feasible, and methods such as theirs may be needed. I think it's important to at least have a discussion on this in the paper, and to be explicit about the difficulty of applying grid search with non-binary attributes.

To Reviewer C1h7's question on why the method of Agarwal et al. may *fail* with non-negative costs, I don't think the authors provided a response. I can see why the surrogate relaxations would become non-convex with negative costs, but given that training of deep networks would anyway result in a non-convex problem, would the surrogate being non-convex be all that problematic? As the reviewer suggested, I think it's important to elaborate on why their method may *fail* (which I think is a strong claim).

Finally, a couple of minor comments on the writing from a quick glance:
- In the abstract, you mention that "We propose an easy-to-use and robust way of imposing fairness constraints when training": are you referring to Lagrangian formulation, which I think has already been well-studied in the literature and not necessarily a contribution of this paper, or specifically to the use of grid-search to tune $\lambda$, which unfortunately is not scalable to multiple constraints / protected groups. It might be good to be a bit more specific about what the paper claims to be its proposal.
- In the intro, you mention that "We also provide some of the first empirical results on fair relaxations for large-scale image and
text classification." Is this statement entirely accurate? I do remember CelebA being used commonly in prior fairness papers, e.g. https://arxiv.org/pdf/2004.01355.pdf.

I trust that the authors will put in the time and effort to make a thorough pass over the paper to polish up the writing, and to ensure that the statements made are all precise and accurate.